# FOCUSING BY CONTRASTIVE ATTENTION: ENHANCING VLMS' VISUAL REASONING

## ABSTRACT

Vision-Language Models (VLMs) have demonstrated remarkable success across diverse visual tasks, yet their performance degrades in complex visual environments. While existing enhancement approaches require additional training, rely on external segmentation tools, or operate at coarse-grained levels, they overlook the innate ability within VLMs. To bridge this gap, we investigate VLMs' attention patterns and discover that: (1) visual complexity strongly correlates with attention entropy, negatively impacting reasoning performance; (2) attention progressively refines from global scanning in shallow layers to focused convergence in deeper layers, with convergence degree determined by visual complexity. (3) Theoretically, we prove that the contrast of attention maps between general queries and task-specific queries enables the decomposition of visual signal into semantic signals and visual noise components. Building on these insights, we propose **C**ontrastive **A**ttention **R**efinement for **V**isual **E**nhancement (**CARVE**), a training-free method that extracts task-relevant visual signals through attention contrasting at the pixel level. Extensive experiments demonstrate that CARVE consistently enhances performance, achieving up to 75% improvement on open-source models. Our work provides critical insights into the interplay between visual complexity and attention mechanisms, offering an efficient pathway for improving visual reasoning with contrasting attention.

## 1 INTRODUCTION

Vision-Language Models (VLMs) have achieved remarkable success across diverse tasks (Radford et al., 2021; Jia et al., 2021; Alayrac et al., 2022). However, in human vision, complex visual features frequently divert attention from task-relevant regions (Treisman & Gelade, 1980). Given this cognitive parallel, a question arises: *Similarly, do complex images interfere with VLMs' attention mechanisms, making it difficult for them to focus on task-relevant regions?*

To answer this question, we investigate the relationship between visual complexity and attention patterns via quantitative experiments. Specifically, we define visual complexity as texture and color dimensions, revealing a significant positive correlation between both factors and attention entropy. Furthermore, our analysis shows that attention entropy negatively correlates with accuracy on visual reasoning tasks. Through this two-stage analysis, we establish that **complex visual information impairs VLMs' reasoning performance via attention distribution** (detailed in Section 3).

Based on these findings, we conduct a preliminary experiment on TextVQA (Singh et al., 2019) by first applying progressive masking to obscure background regions, then cropping to retain only task-relevant regions and adaptively magnifying them to the original image size. Figure 1 presents two representative samples where cluttered visual environments initially cause incorrect predictions. While incorrect token probability initially prevails in both samples, correct token probability surpasses incorrect probability at mask ratios of approximately 0.02 and 0.65 respectively. These results provide initial validation that **masking visual noise can improve correct token probability**.

To automate the visual noise masking process, we leverage contrasting attention maps between general instructions and task-specific questions to distinguish semantic signal from visual noise. To this end, we propose **C**ontrastive **A**ttention **R**efinement for **V**isual **E**nhancement (**CARVE**), a contrastive method for visual extraction. By masking with contrastive attention maps, CARVE crops and magnifies semantic regions to focus on essential semantic signal (detailed in Section 4).

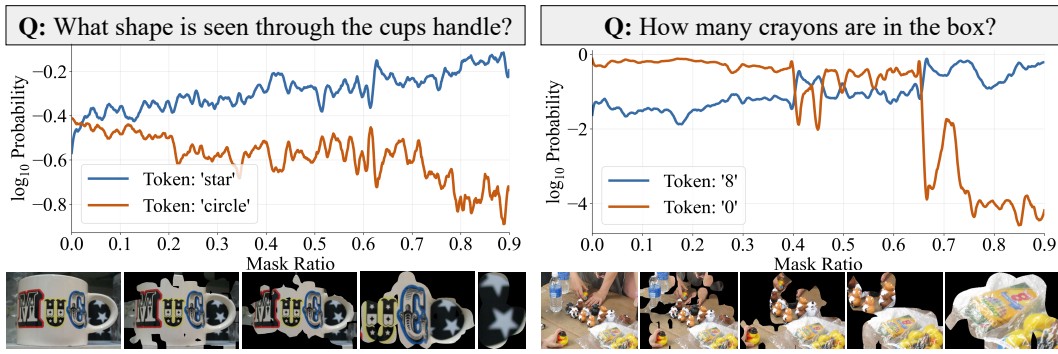

Figure 1: The effect of manually progressive masking on candidate tokens probabilities predicted by QWEN2.5-VL-3B. The x-axis represents mask ratio and y-axis shows $\log_{10}$ probability.

## 2 RELATED WORK

**Contrastive Learning in LLMs.** Liu et al. (2023b) pioneered contrastive objectives for aligning LLMs with human preferences, establishing the foundation for RECIPE (Chen et al., 2024b), which trains a Knowledge Sentinel to determine when queries trigger knowledge updates. Building on alignment challenges, Jiang et al. (2024) employ hallucinated text as hard negatives while Zhang et al. (2024b) apply contrastive learning in hidden representations to suppress hallucinations. Zhai et al. (2025) traces critical transmission paths across all layers, treating less important pathways as negatives, which Pan et al. (2024) further adapted to multimodal LLMs through UniKE's semantic-truthfulness space disentanglement. Departing from embedding-space modifications, DeCK (Bi et al., 2025a) shifts contrastive logic to the decoding stage by comparing token probabilities with and without injected knowledge, while parallel applications emerged in DistiLLM-2 (Ko et al., 2025) for knowledge distillation and Zhu et al. (2024) for factual consistency enhancement.

**Attention-based LLM Optimization.** Alayrac et al. (2022) established multimodal foundations through perceiver resampler architecture, which Li et al. (2023a) refined via a lightweight Querying Transformer for parameter-efficient visual extraction. Extending attention optimization to text modality, Chen et al. (2025) exploit attention scores for dynamic prompt compression through importance sampling at both token and sentence levels. Ma et al. (2024a) eliminated redundant visual token computations while Acharya et al. (2024) introduced block-sparse mechanisms for parallel processing, and Liu et al. (2025b) bypassed attention bottlenecks through sequential chunk processing in Recurrent LLMs. Yao et al. (2025b) identified position bias in multimodal RAG where models over-focus on boundary items. Zhang et al. (2025) discovered that models consistently know where to look, even when they provide the wrong answer. Our method eliminates visual noise by contrasting attention maps to distinguish semantic pixels from noise pixels without requiring training.

## 3 FAILURE TO FOCUS: PHENOMENON, MECHANISM, AND CONSEQUENCE

### 3.1 PHENOMENON: UNDER WHAT CONDITIONS VISUAL FOCUS FAILS

Building upon the question in Section 1, we aim to investigate the underlying causes of VLMs' answering failures. We conduct experiments on TextVQA dataset using QWEN2.5-VL-3B-INSTRUCT. As shown in Figure 2, we visualize attention maps during inference and find two interesting phenomena. Attention progressively refines from broad global scanning in shallow layers to regional localization in the middle layers, culminating in focused convergence in deep layers. The degree of convergence varies based on input images.

Moreover, visual complexity critically influences attention convergence. In simple scenes with clear targets and minimal distractors, the high-attention regions successfully narrow as layers deepen, aligning with task-relevant regions. Conversely, in complex scenes with rich textures and colors, the high-attention regions still attempt to narrow as layers deepen, yet the resulting attention weights remain more diffused compared to simple scenes. As indicated by the annotation "Confused where to look", this attention dispersion resembles human hesitation when confronting crowded shelves and ultimately manifests as reasoning failures. These observations lead us to formulate a question: *Does*

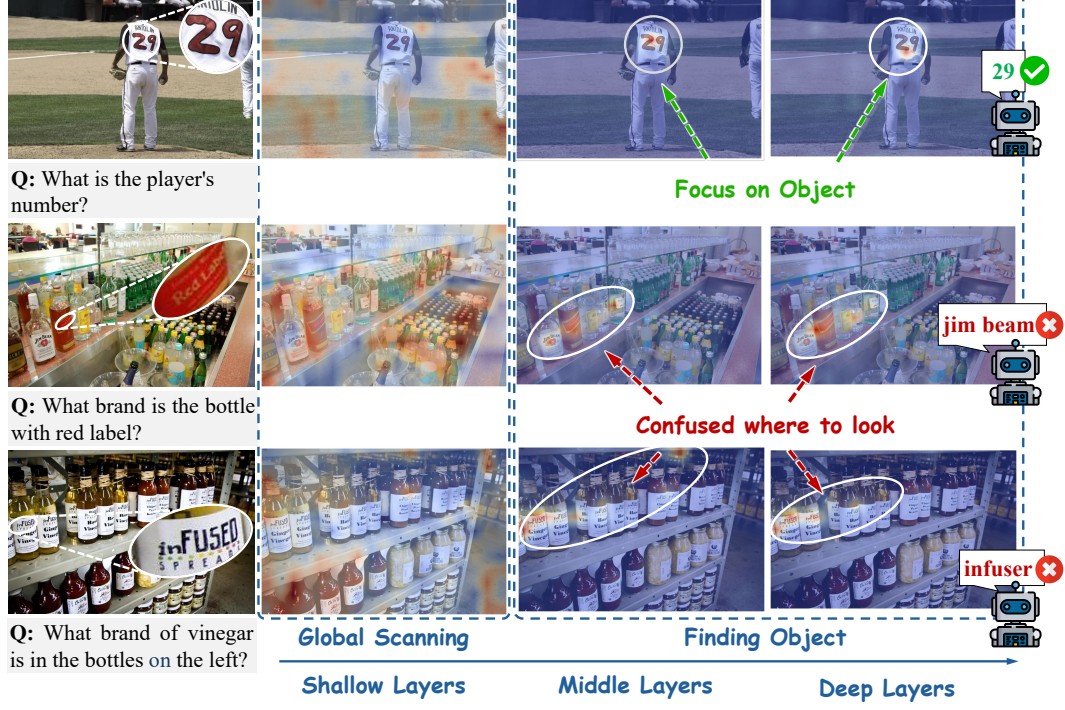

Figure 2: Attention maps across different layers during inference. Each row represents a visual question-answering task: row 1 shows a simple scene with clear targets, while rows 2-3 depict complex scenes with dense textures and multiple similar objects. From left to right, the columns display: input images, shallow layer attention, middle layer attention, and deep layer attention.

*visual complexity affect VLMs' attention distribution, and does this attention distribution further influence VLMs' performance?*

To answer it, we conduct two deeper experiments: First, we quantify the relationship between visual complexity and attention entropy to establish whether complex inputs produce dispersed attention (Section 3.2); Second, we examine the correlation between attention entropy and model performance to determine whether dispersed attention contributes to reasoning failures (Section 3.3).

## 3.2 MECHANISM: THE EFFECT OF VISUAL COMPLEXITY ON ATTENTION ENTROPY

In Figure 2, images in rows 2-3 differ from row 1 by displaying numerous colorful bottles, containing significantly more textures and colors. Therefore, we decompose visual complexity into two dimensions: **texture** and **color**, and investigate their respective impacts on attention.

We define the texture complexity and the color complexity as follows:

**Texture Complexity.** Let $\mathcal{I} \in \mathbb{R}^{H \times W \times 3}$ denote an input image. We define the texture complexity $\mathcal{T}_c(\mathcal{I})$ using Canny edge detection (Canny, 1986), where $\mathcal{E}(\mathcal{I}) \in \{0, 1\}^{H \times W}$ represents the resulting binary edge map. The texture complexity is then defined as:

$$\mathcal{T}_c(\mathcal{I}) = \frac{1}{HW} \sum_{i=1}^{H} \sum_{j=1}^{W} \mathcal{E}(\mathcal{I})_{ij} = \frac{\|\mathcal{E}(\mathcal{I})\|_1}{HW} \in [0, 1] \qquad (3.1)$$

**Color Complexity.** Let $\zeta_{ij} = \text{Hue}(\Psi_{RGB \to HSV}(\mathcal{I}_{ij}))$ denote the hue value at pixel $(i, j)$ after applying the RGB to HSV transformation operator $\Psi$. The color complexity is then defined as:

$$\mathcal{C}_c(\mathcal{I}) = -\frac{1}{\ln B} \sum_{b=0}^{B-1} \rho_b \ln \rho_b, \quad \text{where } \rho_b = \frac{n_b}{HW}, \quad n_b = |\{(i, j) : \zeta_{ij} = b\}| \qquad (3.2)$$

with $B = 180$ hue bins, yielding $\mathcal{C}_c \in [0, 1]$ where higher values indicate greater color diversity.

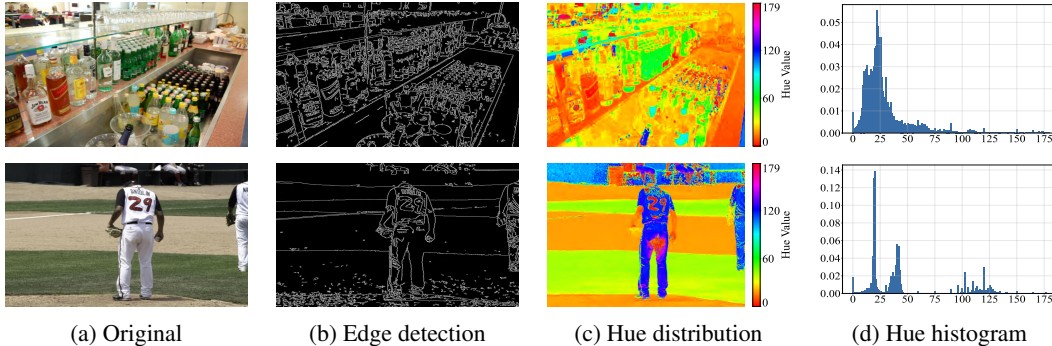

(a) Original          (b) Edge detection          (c) Hue distribution          (d) Hue histogram

Figure 3: Visualization of texture and color complexity analysis. Each row represents a sample image: (a) Original image $\mathcal{I}$, (b) Canny edge map $\mathcal{E}(\mathcal{I})$ for texture complexity $\mathcal{T}_c$, (c) Spatial hue distribution $\zeta$ in HSV space, and (d) Hue statistic (x-axis: hue value, y-axis: ratio).

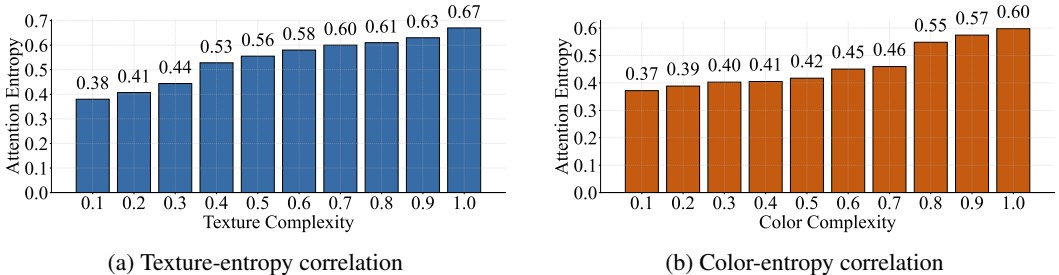

(a) Texture-entropy correlation          (b) Color-entropy correlation

Figure 4: Correlation analysis between visual complexity and attention entropy. Both attention entropy and complexity are normalized to the $[0, 1]$, divided into intervals of 0.1, and the average attention entropy is calculated within each interval.

Figure 3 visually validates the effectiveness of our complexity measurement approach. The first row demonstrates high texture complexity with dense edge networks in $\mathcal{E}(\mathcal{I})$ and diverse color distribution across the hue spectrum. In contrast, the second row exhibits minimal edge density and concentrated hue values, indicating lower complexity scores. We therefore proceed to quantitatively investigate the correlation between complexity metrics and attention distribution.

For measuring the distribution of attention across visual tokens, inspired by Yao et al. (2025b), we employ Shannon entropy (Shannon, 1948) as our quantification metric. Let $N_v$ denote the number of visual tokens in the model's representation. We denote the attention map as $A_{l,t}^{(Q)} \in \mathbb{R}^{N_v}$, where $l$ indicates the layer index, $t$ the generation time step, and $Q$ the input question. For entropy analysis, we focus on the final generation step $t_{\text{end}}$ and define the overall attention entropy $\overline{\mathcal{H}}$ as:

$$\overline{\mathcal{H}} = \frac{1}{|\mathcal{L}|} \sum_{l \in \mathcal{L}} \mathcal{H}(A_{l,t_{\text{end}}}^{(Q)}) = \frac{1}{|\mathcal{L}|} \sum_{l \in \mathcal{L}} \left( -\sum_{i=1}^{N_v} a_{l,t_{\text{end}},i} \ln a_{l,t_{\text{end}},i} \right) \qquad (3.3)$$

where $\mathcal{L} = [L_{\text{start}}, L_{\text{end}}]$ represents the layer range under consideration, and $a_{l,t,i}$ denotes the contrasted attention weight for the $i$-th visual token. Higher entropy indicates more dispersed attention, while lower entropy indicates more concentrated focus.

Figure 4 presents the correlation analysis between our defined complexity metrics and computed attention entropy. Both texture complexity (Figure 4a) and color complexity (Figure 4b) exhibit strong positive linear relationships with attention entropy. This monotonic trend indicates that **complex visual features lead to dispersed attention patterns in VLMs**.

### 3.3 Consequence: How Dispersed Attention Impairs Performance

Figure 5(a) reveals a strong negative correlation between attention entropy and accuracy. As attention entropy increases from 5.1 to 6.8, performance decreases from approximately 76% to 65%, confirming that increased attention dispersion directly impairs visual reasoning capabilities in VLMs.

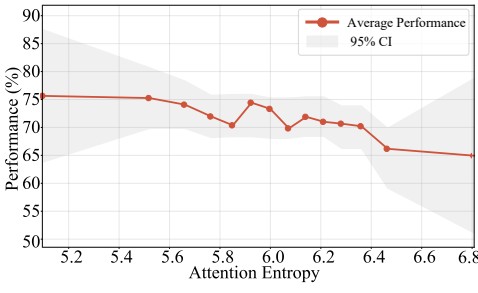 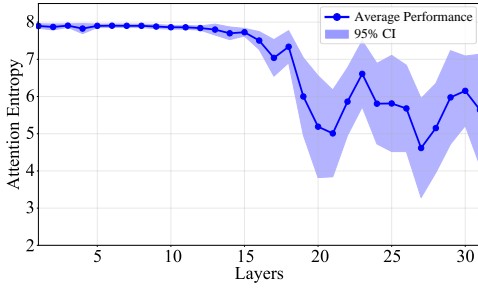

(a) Entropy-accuracy correlation

(b) Layer-wise entropy correlation

Figure 5: Attention entropy's correlation with accuracy and its evolution across layers. Shaded regions indicate 95% confidence intervals computed as $\bar{x} \pm t_{0.975,n-1} \cdot s/\sqrt{n}$. (a) Shows accuracy for samples grouped by overall attention entropy $\overline{\mathcal{H}}$. (b) Displays mean entropy across $N$ samples per layer as $\frac{1}{N}\sum_{i=1}^{N} \mathcal{H}(A_{l,t_{\text{end}}}^{(Q,i)})$, where $A_{l,t_{\text{end}}}^{(Q,i)}$ is sample $i$'s attention at the final generation step.

To investigate the hierarchical evolution of attention entropy, we present mean entropy and its distribution across layers in Figure 5(b). The results reveal two notable characteristics: (1) attention entropy monotonically decreases with layer depth, consistent with Figure 2. (2) The 95% confidence intervals progressively widen with increasing depth, indicating enhanced inter-sample variability. For samples with clear visual targets, deep layers achieve highly concentrated attention. In contrast, for noisy samples, the model maintains dispersed attention patterns even in deep layers.

# 4 CONTRASTIVE ATTENTION REFINEMENT FOR VISUAL ENHANCEMENT

## 4.1 THEORETICAL FOUNDATION: NOISE SUPPRESSION AND VISUAL REFINEMENT

Based on our findings in Section 3, where we demonstrated that visual complexity causes attention dispersion and performance degradation, we seek to extract pure task-related semantic signal. Therefore, we first formally define the attention signal decomposition mechanism.

**Definition 1 (Attention Decomposition):** Attention distributions are influenced by inherent visual noise (detailed in Appendix A.2) of the image and task-related semantic signal. The attention map $A_{l,t}^{(Q)}(\mathcal{I})$ decomposes as:

$$A_{l,t}^{(Q)}(\mathcal{I}) = \mathcal{F}_{\text{vis}}(\mathcal{I}) \otimes \mathcal{F}_{\text{sem}}(Q, \mathcal{I}) \tag{4.1}$$

where $\mathcal{F}_{\text{vis}}(\mathcal{I}) \in \mathbb{R}^{N_v}$ captures image-inherent visual noise, $\mathcal{F}_{\text{sem}}(Q, \mathcal{I}) \in \mathbb{R}^{N_v}$ captures task-related semantic signal, and $\otimes$ denotes the Hadamard product.

When using general instructions $G$, due to the absence of specific tasks to introduce semantic information, the semantic signal function reduces to uniform distribution ($\mathcal{F}_{\text{sem}}(G, \mathcal{I}) \approx \mathbf{1}_{N_v}$), making general instruction attention predominantly capture visual noise:

$$A_{l,t}^{(G)}(\mathcal{I}) \approx \mathcal{F}_{\text{vis}}(\mathcal{I}) \otimes \mathbf{1}_{N_v} = \mathcal{F}_{\text{vis}}(\mathcal{I}) \tag{4.2}$$

**Definition 2 (Semantic Extraction Based on Attention Decomposition):** To extract semantic signal function $\mathcal{F}_{\text{sem}}(Q, \mathcal{I})$ from $A^{(Q)}$, we define estimated semantic attention $\hat{A} \in \mathbb{R}_+^{N_v}$ as our estimate of $\mathcal{F}_{\text{sem}}(Q, \mathcal{I})$, which is the solution to the following optimization problem:

$$\hat{A} = \arg\min_{\tilde{A} \in \mathcal{A}} \mathcal{J}(\tilde{A}; A^{(Q)}, A^{(G)}) \tag{4.3}$$

where the objective function is constructed based on Definition 1's decomposition:

$$\mathcal{J}(\tilde{A}) = \underbrace{\sum_{i=1}^{N_v} \left( \tilde{A}_i \cdot \mathcal{F}_{\text{vis},i}(\mathcal{I}) - [\mathcal{F}_{\text{vis},i}(\mathcal{I}) \cdot \mathcal{F}_{\text{sem},i}(Q, \mathcal{I})] \right)^2}_{\text{Semantic reconstruction error}} + \underbrace{\lambda \sum_{i=1}^{N_v} \tilde{A}_i^2 \cdot \mathcal{F}_{\text{vis},i}(\mathcal{I})}_{\text{Visual suppression regularization}} \tag{4.4}$$

Figure 6: CARVE comprises three stages: Stage 1 generates general attention distribution $A_i^{(G)}$ with general instructions; Stage 2 extracts task-specific attention $A_i^{(Q)}$; Stage 3 applies contrasted attention $\hat{A}_i$ to generate enhanced masked images for noise suppression.

**Theorem 3 (Closed-form Solution for Semantic Extraction):** Substituting Definition 1's relationships $A_i^{(Q)} \approx \mathcal{F}_{\text{vis},i} \cdot \mathcal{F}_{\text{sem},i}$ and $A_i^{(G)} \approx \mathcal{F}_{\text{vis},i}$ into the optimization objective yields:

$$\mathcal{J}(\tilde{A}) = \sum_{i=1}^{N_v} \left( \tilde{A}_i \cdot A_i^{(G)} - A_i^{(Q)} \right)^2 + \lambda \sum_{i=1}^{N_v} \tilde{A}_i^2 \cdot A_i^{(G)} \tag{4.5}$$

where $\lambda > 0$ is a regularization parameter that controls the strength of visual noise suppression.

Solving the first-order optimality conditions yields the closed-form solution:

$$\hat{A}_i = \frac{A_i^{(Q)}}{A_i^{(G)} + \lambda} = \frac{\mathcal{F}_{\text{vis},i} \cdot \mathcal{F}_{\text{sem},i}}{\mathcal{F}_{\text{vis},i} + \lambda} \approx \mathcal{F}_{\text{sem},i} \quad \text{when } \mathcal{F}_{\text{vis},i} \gg \lambda \tag{4.6}$$

Equation 4.6 demonstrates that normalization suppresses the influence of $\mathcal{F}_{\text{vis},i}$ when it dominates (i.e., $\mathcal{F}_{\text{vis},i} \gg \lambda$), approximating the semantic signal $\mathcal{F}_{\text{sem},i}$ (detailed analysis in Appendix C).

### 4.2 CONTRASTIVE ATTENTION-BASED VISUAL ENHANCEMENT

Having obtained the semantically refined attention maps $\{\hat{A}\}$, as shown in Algorithm 1, we now proceed to generate attention masks that physically remove visual noise from the input image.

**Attention Maps Fusion.** Since different layers and time steps capture complementary information, we fuse attention maps across the layer range $\mathcal{L}$ and generation time steps $\mathcal{T} = [t_{\text{start}}, t_{\text{end}}]$ through weighted aggregation. Later tokens encode richer contextual information by accessing complete preceding sequences during inference, thus receiving higher fusion weights.

**Mask Generation and Visual Extraction.** Task-relevant regions are identified by applying the top-p percentile threshold $\tau = \mathcal{Q}_p(S)$, which retains the top $p \in (0,1]$ proportion of pixels from attention map $S$. Connected component analysis extracts coherent regions from the thresholded map. We select the top-$K$ regions ranked by cumulative attention scores and generate the enhanced image through $\mathcal{I}_{\text{refined}} = \Phi(\mathcal{I}, M^*)$, where $\Phi$ applies masking, cropping, and resizing, and $K$ controls the maximum number of regions to preserve. This refinement eliminates visual noise while magnifying task-relevant content, enabling focused attention on task-related areas.

| Model | Step: $\mathcal{T}$ | A-OKVQA | POPE | V* | TextVQA |
|---|---|---|---|---|---|
| QWEN2.5-VL-3B | w/o CARVE | $73.0_{(-)}$ | $86.9_{(-)}$ | $50.3_{(-)}$ | $72.8_{(-)}$ |
| | $t_{\text{start}}$ | $76.5_{(\uparrow 4.79)}$ | $87.1_{(\uparrow 0.23)}$ | $56.0_{(\uparrow 11.33)}$ | $76.1_{(\uparrow 4.53)}$ |
| | $t_{\text{end}}$ | $\mathbf{79.2}_{(\uparrow 8.49)}$ | $\mathbf{88.4}_{(\uparrow 1.73)}$ | $\mathbf{57.1}_{(\uparrow 13.52)}$ | $\mathbf{76.4}_{(\uparrow 4.95)}$ |
| | $\mathcal{T}_{\text{full}}$ | $78.3_{(\uparrow 7.26)}$ | $87.9_{(\uparrow 1.15)}$ | $56.5_{(\uparrow 12.33)}$ | $76.3_{(\uparrow 4.81)}$ |
| QWEN2.5-VL-7B | w/o CARVE | $75.0_{(-)}$ | $87.0_{(-)}$ | $50.8_{(-)}$ | $75.0_{(-)}$ |
| | $t_{\text{start}}$ | $77.0_{(\uparrow 2.67)}$ | $87.9_{(\uparrow 1.03)}$ | $58.6_{(\uparrow 15.35)}$ | $80.7_{(\uparrow 7.60)}$ |
| | $t_{\text{end}}$ | $\mathbf{78.3}_{(\uparrow 4.40)}$ | $\mathbf{89.7}_{(\uparrow 3.10)}$ | $\mathbf{59.7}_{(\uparrow 17.52)}$ | $\mathbf{81.9}_{(\uparrow 9.20)}$ |
| | $\mathcal{T}_{\text{full}}$ | $78.0_{(\uparrow 4.00)}$ | $88.6_{(\uparrow 1.84)}$ | $58.1_{(\uparrow 14.37)}$ | $81.7_{(\uparrow 8.93)}$ |
| LLAVA1.5-7B | w/o CARVE | $71.5_{(-)}$ | $83.6_{(-)}$ | $38.7_{(-)}$ | $47.8_{(-)}$ |
| | $t_{\text{start}}$ | $73.9_{(\uparrow 3.36)}$ | $86.8_{(\uparrow 3.83)}$ | $57.1_{(\uparrow 47.55)}$ | $57.9_{(\uparrow 21.13)}$ |
| | $t_{\text{end}}$ | $\mathbf{78.2}_{(\uparrow 9.37)}$ | $\mathbf{89.0}_{(\uparrow 6.46)}$ | $\mathbf{66.5}_{(\uparrow 71.83)}$ | $\mathbf{58.2}_{(\uparrow 21.76)}$ |
| | $\mathcal{T}_{\text{full}}$ | $75.4_{(\uparrow 5.45)}$ | $89.0_{(\uparrow 6.46)}$ | $66.5_{(\uparrow 71.83)}$ | $57.9_{(\uparrow 21.13)}$ |
| LLAVA1.5-13B | w/o CARVE | $75.7_{(-)}$ | $84.6_{(-)}$ | $42.4_{(-)}$ | $57.1_{(-)}$ |
| | $t_{\text{start}}$ | $76.2_{(\uparrow 0.66)}$ | $90.0_{(\uparrow 6.38)}$ | $65.4_{(\uparrow 54.25)}$ | $59.2_{(\uparrow 3.68)}$ |
| | $t_{\text{end}}$ | $\mathbf{76.9}_{(\uparrow 1.59)}$ | $\mathbf{90.7}_{(\uparrow 7.21)}$ | $\mathbf{74.3}_{(\uparrow 75.24)}$ | $\mathbf{61.2}_{(\uparrow 7.18)}$ |
| | $\mathcal{T}_{\text{full}}$ | $76.5_{(\uparrow 1.06)}$ | $90.1_{(\uparrow 6.50)}$ | $70.0_{(\uparrow 65.09)}$ | $\mathbf{61.2}_{(\uparrow 7.18)}$ |

Table 1: Accuracy comparison of CARVE across VLMs on four datasets. We evaluate three temporal configurations: $t_{\text{start}}$ uses attention from initial generated tokens, $t_{\text{end}}$ from final tokens, and $\mathcal{T}_{\text{full}}$ applies weighted fusion across all tokens. We use layer range $\mathcal{L} = [20, 25]$ for attention fusion.

---

**Algorithm 1** CARVE: Contrastive Attention Refinement for Visual Enhancement

---

**Notation:** $\mathcal{M}$: VLM model; $\Xi$: attention extraction; $\pi_{H \times W}$: spatial reshape; $\mathcal{Q}_p$: top-$p$ threshold; $\Phi$: visual extraction (mask, crop, resize); $G$: general instruction; $\tau$: threshold; $\mathcal{R}$: connected regions; $K$: max regions to keep

**Require:** $\mathcal{I} \in \mathbb{R}^{H \times W \times 3}$, $Q$, $\mathcal{M}$, $\Theta = \{\mathcal{L}, \mathcal{T}, p, \lambda, K\}$

1: **Inference:** $\mathcal{A}^Q \leftarrow \{A_{l,t}^{(Q)}\}_{l \in \mathcal{L}, t \in \mathcal{T}} = \Xi(\mathcal{M}, \mathcal{I}, Q)$ $\qquad\qquad$ ▷ Question-specific attention

2: **Inference:** $\mathcal{A}^G \leftarrow \{A_{l,t}^{(G)}\}_{l \in \mathcal{L}, t \in \mathcal{T}} = \Xi(\mathcal{M}, \mathcal{I}, G)$ $\qquad\qquad\qquad$ ▷ General attention

3: **Contrast:** $\hat{A}_{l,t} \leftarrow \dfrac{A_{l,t}^{(Q)}}{A_{l,t}^{(G)} + \lambda}$ for all $l \in \mathcal{L}, t \in \mathcal{T}$ $\qquad\qquad$ ▷ following Eq. 4.6

4: **Fuse:** $S \leftarrow \sum_{t \in \mathcal{T}} w_t \sum_{l \in \mathcal{L}} \pi_{H \times W}(\hat{A}_{l,t})$, $w_t = t - t_{\text{start}} + 1$ $\quad$ ▷ Weighted attention fusion

5: **Threshold:** $\tau \leftarrow \mathcal{Q}_p(S)$ $\qquad\qquad\qquad$ ▷ Compute threshold to retain top $p$ percentile

6: **Mask:** $M^* = \bigcup_{k=1}^{K} R_k^*$ where $R_k^* = \arg\max_{R \in \mathcal{R}} \sum_{(i,j) \in R} S(i,j)$ with $\mathcal{R}$ from $S \geq \tau$

7: **Extract:** $\mathcal{I}_{\text{refined}} \leftarrow \Phi(\mathcal{I}, M^*)$ $\qquad\qquad\qquad\qquad\qquad$ ▷ Visual extraction

8: **Inference: return** $\mathcal{M}(\mathcal{I}_{\text{refined}}, Q)$ $\qquad\qquad\qquad\qquad\qquad$ ▷ Final inference

---

As shown in Figure 6, we propose **C**ontrastive **A**ttention **R**efinement for **V**isual **E**nhancement (**CARVE**), a method that contrasts attention maps to distinguish semantic pixels from noise, preserving only task-relevant regions for enhanced model focus (detailed in Appendix B).

## 5 METHOD ANALYSIS

### 5.1 EXPERIMENTAL SETUP

**Datasets.** We conduct our experiments on four datasets: A-OKVQA (Schwenk et al., 2022), POPE (Li et al., 2023b), V* (Wu & Xie, 2023), and TextVQA (Singh et al., 2019), which cover multiple task dimensions including visual reasoning, visual understanding, and visual knowledge reasoning. For TextVQA, we evaluate the models' intrinsic visual text recognition capabilities by providing only images and questions without external OCR augmentation (detailed in Appendix E).

**Models.** We conduct experiments on four VLMs: QWEN2.5-VL-3B-INSTRUCT, QWEN2.5-VL-7B-INSTRUCT (Qwen, 2025), LLAVA-1.5-7B, and LLAVA-1.5-13B (Liu et al., 2023a). The Qwen family processes images at $448 \times 448$ resolution, while the LLaVA-1.5 family operates at $336 \times 336$ resolution. All models employ greedy decoding.

| Model | Layer(s): $\mathcal{L}$ | | A-OKVQA | POPE | V* | TextVQA |
|---|---|---|---|---|---|---|
| QWEN2.5-VL-3B | w/o CARVE | | $73.0_{(-)}$ | $86.9_{(-)}$ | $50.3_{(-)}$ | $72.8_{(-)}$ |
| | Single Layer | 14 | $74.3_{(\uparrow1.78)}$ | $87.1_{(\uparrow0.23)}$ | $53.9_{(\uparrow7.16)}$ | $73.6_{(\uparrow1.10)}$ |
| | | 20 | $76.5_{(\uparrow4.79)}$ | $87.4_{(\uparrow0.58)}$ | $56.0_{(\uparrow11.33)}$ | $74.7_{(\uparrow2.61)}$ |
| | | 25 | $76.7_{(\uparrow5.07)}$ | $87.5_{(\uparrow0.69)}$ | $56.0_{(\uparrow11.33)}$ | $75.9_{(\uparrow4.26)}$ |
| | Multi-Layers | [10, 15] | $74.0_{(\uparrow1.37)}$ | $86.9_{(0.00)}$ | $53.4_{(\uparrow6.16)}$ | $73.0_{(\uparrow0.27)}$ |
| | | [15, 20] | $76.8_{(\uparrow5.21)}$ | $87.7_{(\uparrow0.92)}$ | $56.0_{(\uparrow11.33)}$ | $76.0_{(\uparrow4.40)}$ |
| | | [20, 25] | $\mathbf{78.3}_{(\uparrow7.26)}$ | $\mathbf{87.9}_{(\uparrow1.15)}$ | $\mathbf{57.1}_{(\uparrow13.52)}$ | $\mathbf{76.3}_{(\uparrow4.81)}$ |
| QWEN2.5-VL-7B | w/o CARVE | | $75.0_{(-)}$ | $87.0_{(-)}$ | $50.8_{(-)}$ | $75.0_{(-)}$ |
| | Single Layer | 14 | $75.2_{(\uparrow0.27)}$ | $87.5_{(\uparrow0.57)}$ | $54.5_{(\uparrow7.28)}$ | $75.2_{(\uparrow0.27)}$ |
| | | 20 | $76.9_{(\uparrow2.53)}$ | $87.9_{(\uparrow1.03)}$ | $56.5_{(\uparrow11.22)}$ | $77.9_{(\uparrow3.87)}$ |
| | | 25 | $77.0_{(\uparrow2.67)}$ | $88.2_{(\uparrow1.38)}$ | $57.0_{(\uparrow12.20)}$ | $78.4_{(\uparrow4.53)}$ |
| | Multi-Layers | [10, 15] | $75.0_{(0.00)}$ | $87.0_{(0.00)}$ | $51.3_{(\uparrow0.98)}$ | $75.0_{(0.00)}$ |
| | | [15, 20] | $77.1_{(\uparrow2.80)}$ | $88.4_{(\uparrow1.61)}$ | $57.6_{(\uparrow13.39)}$ | $79.5_{(\uparrow6.00)}$ |
| | | [20, 25] | $\mathbf{78.0}_{(\uparrow4.00)}$ | $\mathbf{88.6}_{(\uparrow1.84)}$ | $\mathbf{58.1}_{(\uparrow14.37)}$ | $\mathbf{81.7}_{(\uparrow8.93)}$ |
| LLAVA1.5-7B | w/o CARVE | | $71.5_{(-)}$ | $83.6_{(-)}$ | $38.7_{(-)}$ | $47.8_{(-)}$ |
| | Single Layer | 14 | $71.7_{(\uparrow0.28)}$ | $85.1_{(\uparrow1.79)}$ | $63.4_{(\uparrow63.82)}$ | $54.0_{(\uparrow12.97)}$ |
| | | 20 | $74.0_{(\uparrow3.50)}$ | $87.2_{(\uparrow4.31)}$ | $65.4_{(\uparrow68.99)}$ | $56.1_{(\uparrow17.36)}$ |
| | | 25 | $74.1_{(\uparrow3.64)}$ | $87.1_{(\uparrow4.19)}$ | $65.4_{(\uparrow68.99)}$ | $56.2_{(\uparrow17.57)}$ |
| | Multi-Layers | [10, 15] | $71.5_{(0.00)}$ | $84.5_{(\uparrow1.08)}$ | $48.2_{(\uparrow24.55)}$ | $49.2_{(\uparrow2.93)}$ |
| | | [15, 20] | $74.2_{(\uparrow3.78)}$ | $87.5_{(\uparrow4.67)}$ | $65.4_{(\uparrow68.99)}$ | $56.4_{(\uparrow17.99)}$ |
| | | [20, 25] | $\mathbf{75.4}_{(\uparrow5.45)}$ | $\mathbf{89.0}_{(\uparrow6.46)}$ | $\mathbf{66.5}_{(\uparrow71.83)}$ | $\mathbf{58.2}_{(\uparrow21.76)}$ |
| LLAVA1.5-13B | w/o CARVE | | $75.7_{(-)}$ | $84.6_{(-)}$ | $42.4_{(-)}$ | $57.1_{(-)}$ |
| | Single Layer | 14 | $75.8_{(\uparrow0.13)}$ | $86.1_{(\uparrow1.77)}$ | $66.5_{(\uparrow56.84)}$ | $58.2_{(\uparrow1.93)}$ |
| | | 20 | $76.2_{(\uparrow0.66)}$ | $88.2_{(\uparrow4.26)}$ | $68.6_{(\uparrow61.79)}$ | $59.1_{(\uparrow3.50)}$ |
| | | 25 | $76.2_{(\uparrow0.66)}$ | $88.1_{(\uparrow4.14)}$ | $69.0_{(\uparrow62.74)}$ | $59.2_{(\uparrow3.68)}$ |
| | Multi-Layers | [10, 15] | $75.7_{(0.00)}$ | $85.0_{(\uparrow0.47)}$ | $52.9_{(\uparrow24.76)}$ | $57.4_{(\uparrow0.53)}$ |
| | | [15, 20] | $76.8_{(\uparrow1.45)}$ | $88.6_{(\uparrow4.73)}$ | $69.1_{(\uparrow62.97)}$ | $59.4_{(\uparrow4.03)}$ |
| | | [20, 25] | $\mathbf{76.9}_{(\uparrow1.59)}$ | $\mathbf{90.1}_{(\uparrow6.50)}$ | $\mathbf{70.0}_{(\uparrow65.09)}$ | $\mathbf{61.2}_{(\uparrow7.18)}$ |

Table 2: We investigate CARVE's accuracy using both single-layer and multi-layer intervention strategies at shallow, middle, and deep model depths, where single-layer interventions use attention maps $\hat{A}_i$ from individual layers, while multi-layer interventions fuse maps across multiple layers to guide masking decisions. We employ $\mathcal{T}_{\text{full}}$ as the time step configuration.

## 5.2 RESULTS

**CARVE Enhances VLMs' Visual QA Performance.** Tables 1 and 2 demonstrate CARVE's consistent performance enhancement across all evaluated models and datasets. Earlier-generation models exhibit substantially greater improvements than their more recent counterparts. For instance, LLAVA1.5-7B achieves a 71.83% relative improvement on V*, whereas QWEN2.5-VL-7B shows a 17.52% gain. This pattern indicates that limited-capability models suffer more from visual complexity interference and benefit more from contrastive attention-guided focusing mechanisms.

**Ablation Study on the Time Step.** Table 1 reveals a consistent performance hierarchy across various time step selection strategies. Specifically, $t_{\text{end}}$ generally outperforms $\mathcal{T}_{\text{full}}$, which in turn surpasses $t_{\text{start}}$ across most experimental configurations. This pattern is exemplified by QWEN2.5-VL-7B's performance on TextVQA, where $t_{\text{end}}$ achieves 81.9% accuracy, followed by $\mathcal{T}_{\text{full}}$ at 81.7% and $t_{\text{start}}$ at 80.7%. This phenomenon aligns with architectural principles. Later tokens encode richer contextual information by accessing complete preceding sequences during inference. Consequently, the final token's attention maps accurately localize target objects, providing prerequisite conditions for CARVE's noise masking mechanism.

**Ablation Study on the Layer Selection.** To investigate layer selection effects on attention pattern extraction, we conduct systematic experiments as shown in Table 2. Across all tested model architectures, the layer-wise performance demonstrates the following general ordering from best to worst: [20,25], [15,20], single layer 25, single layer 20, single layer 14, and [10,15]. This pattern is exemplified by LLAVA1.5-7B's performance on TextVQA, where the multi-layer [20,25] achieves a 21.76% improvement, the [15,20] reaches a 17.99% improvement, while the early-layer [10,15] attains only a 2.93% improvement. Multi-layer fusion outperforms single-layer alternatives by capturing complementary information and providing robustness against individual layer randomness.

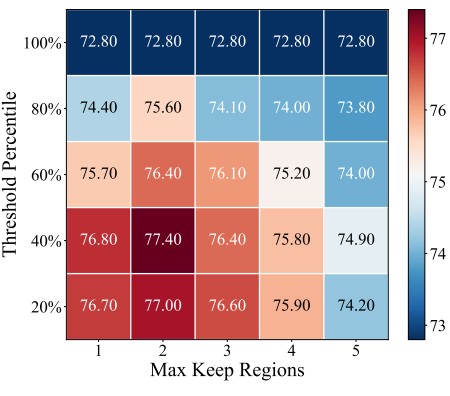 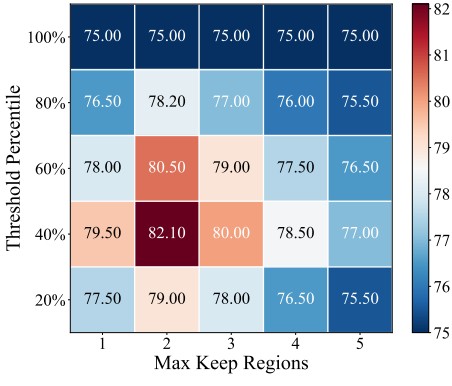

(a) Accuracy for QWEN2.5-VL-3B on TextVQA

(b) Accuracy for QWEN2.5-VL-7B on TextVQA

Figure 7: Impact of mask generation hyperparameters on TextVQA accuracy for QWEN family. Results show performance across varying top-$p$ threshold and maximum keep regions $K$.

| Method | Original | SAM | YOLO | CLIP | ViCrop | | | CARVE |
| --- | --- | --- | --- | --- | --- | --- | --- | --- |
| | | | | | rel-att | grad-att | pure-grad | |
| Accuracy | 47.80 | 49.42 | 48.84 | 48.55 | 55.17 | 56.06 | 51.67 | **58.2** |
| GPU Time | 0.17 | 3.33 | 0.35 | 1.07 | 1.16 | 0.89 | 2.36 | 1.34 |

Table 3: Performance comparison of CARVE against external tool-based approaches and ViCrop on TextVQA: accuracy (%) and inference time overhead per sample (seconds).

This phenomenon aligns with our findings in Figure 5(b): early layers perform global scanning with high entropy, while middle-to-deep layers focus on task-relevant patterns.

**Sensitivity Analysis of Mask Generation.**   We examine a 1,000-instance subset randomly sampled from TextVQA. As shown in Figure 7, when $p = 1.0$, corresponding to no masking intervention, performance remains at original levels. However, when $p$ is set within $[0.2, 0.6]$ combined with $K \in \{2, 3\}$, the model achieves optimal performance, as these settings maintain a balance between preserving object representations and suppressing visual noise. In contrast, aggressive masking strategies manifest detrimental effects: retention ratios set to 20% and single-region constraints lead to degradation, since such aggressive configurations discard essential visual information.

**Comparative Analysis with Alternative Methods.**   As shown in Table 3, CARVE substantially outperforms external tool-based approaches: SAM (Kirillov et al., 2023), YOLO (Redmon et al., 2016), CLIP (Radford et al., 2021) and recent ViCrop (Zhang et al., 2025) variants across diverse baseline methodologies (conducted on NVIDIA RTX A6000). External tools rely on generic segmentation algorithms that lack question-image context awareness. While ViCrop effectively reduces visual noise through strategic cropping, it lacks pixel-level noise masking. Regarding computational efficiency, CARVE requires 1.34 seconds of GPU processing time, exceeding simpler approaches such as YOLO (0.35 seconds) but remaining within practical deployment constraints.

## 6 CONCLUSION

In this work, we demonstrate that visual complexity correlates with attention entropy, which in turn negatively impacts VLMs' performance. Theoretically, we prove that contrasting attention maps between general and specific instructions enables effective decomposition of visual signal into semantic signal and visual noise components. To this end, we propose **C**ontrastive **A**ttention **R**efinement for **V**isual **E**nhancement (**CARVE**), a training-free method that leverages this theoretical insight to extract task-relevant signal through attention contrasting and pixel-level masking. Our work provides critical insights into the interplay between visual complexity and attention mechanisms, offering an efficient pathway for improving visual reasoning without training.

ETHICS STATEMENT

In conducting our research, we prioritize ethical standards to ensure integrity and contribute positively to the scientific community. We exclusively utilize open-source datasets, ensuring our work builds upon accessible and transparent resources. Our methods employ widely recognized models with established reliability within the academic community. We have designed our methodology to prevent generating harmful or misleading information, safeguarding our findings' integrity.

REPRODUCIBILITY STATEMENT

To ensure reproducibility, we utilize publicly available datasets with detailed processing procedures documented in the appendices. Our methodology is fully specified through pseudocode, mathematical formulations, and comprehensive descriptions. All configurations, hyperparameters, and evaluation protocols are explicitly documented. Theoretical contributions include complete proofs with assumptions clearly stated. Code and implementation details will be released upon acceptance.

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

# A  DEFINITION AND EXPLANATION

## A.1  DEFINITION

| Symbol | Definition | Description |
|---|---|---|
| $\mathcal{I} \in \mathbb{R}^{H \times W \times 3}$ | Input image | Image with height $H$ and width $W$ |
| $H, W$ | Image dimensions | Height and width in pixels |
| $Q$ | Task-specific question | Task-specific question |
| $G$ | General instruction | General instruction |
| $A_{l,t}^{(Q)} \in \mathbb{R}^{N_v}$ | Question attention map | Attention map at layer $l$, step $t$ |
| $A_{l,t}^{(G)} \in \mathbb{R}^{N_v}$ | General attention map | General question attention map |
| $N_v$ | Visual tokens | Number of visual tokens |
| $\mathcal{L}$ | Layer range | Layer indices |
| $\mathcal{T}$ | Time range | Generation time step |
| $t_{\text{end}}$ | Final step | Final generation step |
| $\mathcal{H}(\cdot)$ | Shannon entropy | Attention distribution entropy |
| $\overline{\mathcal{H}}$ | Overall entropy | Layer-averaged attention entropy |
| $\mathcal{T}_c(\mathcal{I})$ | Texture complexity | Edge density from Canny detection |
| $\mathcal{C}_c(\mathcal{I})$ | Color complexity | Hue diversity measure |
| $\mathcal{E}(\mathcal{I}) \in \{0,1\}^{H \times W}$ | Edge map | Binary edge map from Canny |
| $\Psi_{RGB \to HSV}$ | Color transform | RGB to HSV transformation operator |
| $\zeta_{ij}$ | Hue value | Hue value at pixel $(i,j)$ |
| $\rho_b$ | Hue proportion | Fraction of pixels in hue bin $b$ |
| $B$ | Hue bins | Number of hue bins |
| $\mathcal{F}_{\text{vis}}(\mathcal{I}) \in \mathbb{R}_+^{N_v}$ | Visual noise factor | Image-inherent visual noise component |
| $\mathcal{F}_{\text{sem}}(Q, \mathcal{I}) \in \mathbb{R}_+^{N_v}$ | Semantic signal factor | Task-related semantic signal component |
| $\mathbf{1}_{N_v}$ | Uniform vector | Vector of ones |
| $\hat{A} \in \mathbb{R}_+^{N_v}$ | Estimated attention | Estimated semantic attention |
| $\lambda > 0$ | Regularization | Regularization parameter |
| $p \in (0,1]$ | Top-p percentile | Percentile threshold for masking |
| $K \in \mathbb{N}$ | Max regions | Maximum regions to preserve |
| $w_t$ | Temporal weights | Later token weighting with $w_t = t - t_{\text{start}} + 1$ |
| $S \in \mathbb{R}^{H \times W}$ | Fused map | Spatially reshaped attention map |
| $\mathcal{Q}_p(\cdot)$ | Percentile function | Top-$p$ percentile operator |
| $\tau$ | Threshold | Computed threshold value |
| $M^* \subseteq \{1..H\} \times \{1..W\}$ | Final mask | Union of top-$K$ regions |
| $R_k$ | Connected region | Connected component from thresholding |
| $\Phi(\mathcal{I}, M)$ | Visual extraction | Masking, cropping and resizing |
| $\pi_{H \times W}$ | Spatial reshape | Token to image projection |
| $\Xi$ | Attention extractor | Function to extract attention maps |
| $\mathcal{M}$ | VLM | Vision-language model |
| $L_{\text{total}}$ | Total layers | Number of model layers |
| $N_q$ | Text tokens | Number of query tokens |

## A.2  EXPLANATION

- **Time Step ($t$):** In the autoregressive generation process of vision-language models, a time step denotes the sequential position index in the output token sequence. The model generates responses token-by-token, where $t = 1$ corresponds to the first generated token and $t = t_{\text{end}}$ represents the final token. At each time step, the model produces an attention distribution $A_{l,t}^{(Q)} \in \mathbb{R}^{N_v}$ over visual tokens.

- **Visual Complexity ($\mathcal{T}_c(\mathcal{I})$, $\mathcal{C}_c(\mathcal{I})$):** Visual complexity quantifies the inherent characteristics of an image that can interfere with VLMs' attention mechanisms, decomposed into two orthogonal dimensions. Texture complexity $\mathcal{T}_c(\mathcal{I}) \in [0,1]$ measures the density of edge information using Canny edge detection, where higher values indicate more intricate patterns, object boundaries, and structural details. Color complexity $\mathcal{C}_c(\mathcal{I}) \in [0,1]$ captures the diversity of hue distribution in HSV color space through Shannon entropy, where higher values reflect greater chromatic variation.

- **Visual Tokens ($N_v$):** Visual tokens constitute the discrete representational units obtained after processing an input image through a visual encoder. An image of dimensions $H \times W$ is partitioned and encoded into $N_v$ visual tokens, which form the fundamental units for visual information processing. The attention mechanism allocates weights across these $N_v$ tokens to determine which image regions to attend to.

- **Semantic Signal Factor ($\mathcal{F}_{\mathbf{sem}}(Q, \mathcal{I})$):** The semantic signal factor represents the question-specific component in the attention decomposition framework, valued in $\mathbb{R}_+^{N_v}$. This factor quantifies the semantic signal between each visual token and the given question $Q$. Under general instructions $G$ (e.g., "describe this image"), this factor approximates a uniform distribution ($\mathcal{F}_{\mathrm{sem}}(G, \mathcal{I}) \approx \mathbf{1}_{N_v}$), whereas task-specific questions yield elevated values in semantically relevant regions.

- **Visual Noise Factor ($\mathcal{F}_{\mathbf{vis}}(\mathcal{I})$):** The visual noise factor captures the image-inherent, question-independent attention component, valued in $\mathbb{R}_+^{N_v}$. This factor, determined by texture complexity and color diversity of the image, reflects the influence of visual content characteristics on attention distribution. Under general instructions, the attention distribution is predominantly governed by this factor: $A_{l,t}^{(G)}(\mathcal{I}) \approx \mathcal{F}_{\mathrm{vis}}(\mathcal{I})$.

# B IMPLEMENTATION DETAILS

We conduct our experiments on a server with $4 \times$ NVIDIA RTX A6000 GPUs. $\tau$ is set to 0.05. In practical implementation, CARVE requires three inference passes; however, the first two passes (extracting general instruction and task-specific question) can be terminated early. Specifically, when we require attention maps only from layers $\mathcal{L} = [L_{start}, L_{end}]$, the first two inference processes can halt upon completing layer $L_{end}$ computation, eliminating the need for full $L_{\mathrm{total}}$ layer forward propagation. The third inference must run completely to generate the final answer.

# C PROOFS AND ADDITIONAL THEOREMS

## C.1 MATHEMATICAL BASIS OF ATTENTION DECOMPOSITION

**Theorem C.1 (Existence of Attention Decomposition):** For any attention distribution $A_{l,t}^{(Q)}(\mathcal{I}) \in \mathbb{R}_+^{N_v}$, there exists a unique decomposition:

$$A_{l,t}^{(Q)}(\mathcal{I}) = \mathcal{F}_{\mathrm{vis}}(\mathcal{I}) \otimes \mathcal{F}_{\mathrm{sem}}(Q, \mathcal{I}) \tag{C.1}$$

*Proof:* Define a logarithmic space mapping $\phi : \mathbb{R}_+ \to \mathbb{R}$ where $\phi(x) = \log(x)$. Under this transformation, the decomposition becomes additive in logarithmic space:

$$\phi(A_{l,t}^{(Q)}(\mathcal{I})) = \phi(\mathcal{F}_{\mathrm{vis}}(\mathcal{I})) + \phi(\mathcal{F}_{\mathrm{sem}}(Q, \mathcal{I})) \tag{C.2}$$

Given the boundary condition that $\mathcal{F}_{\mathrm{sem}}(G, \mathcal{I}) = \mathbf{1}_{N_v}$ when $Q = G$ (general instruction), we obtain:

$$\phi(\mathcal{F}_{\mathrm{vis}}(\mathcal{I})) = \phi(A_{l,t}^{(G)}(\mathcal{I})) \tag{C.3}$$

Consequently, through substitution:

$$\phi(\mathcal{F}_{\mathrm{sem}}(Q, \mathcal{I})) = \phi(A_{l,t}^{(Q)}(\mathcal{I})) - \phi(A_{l,t}^{(G)}(\mathcal{I})) \tag{C.4}$$

The unique solution is obtained via the inverse mapping $\phi^{-1}(x) = \exp(x)$. $\square$

## C.2 CONVEXITY ANALYSIS OF THE OPTIMIZATION PROBLEM

**Theorem C.2 (Strict Convexity of Objective Function):** The optimization objective

$$\mathcal{J}(\tilde{A}) = \sum_{i=1}^{N_v} \left( \tilde{A}_i \cdot A_i^{(G)} - A_i^{(Q)} \right)^2 + \lambda \sum_{i=1}^{N_v} \tilde{A}_i^2 \cdot A_i^{(G)} \tag{C.5}$$

is strictly convex with respect to $\tilde{A}$.

*Proof:* Computing the Hessian matrix reveals its structure. Since the objective function is separable across components $\tilde{A}_i$, the Hessian is diagonal with elements:

$$H_{ii} = \frac{\partial^2 \mathcal{J}}{\partial \tilde{A}_i^2} = 2(A_i^{(G)})^2 + 2\lambda A_i^{(G)} = 2A_i^{(G)}(A_i^{(G)} + \lambda) \tag{C.6}$$

Given that $A_i^{(G)} > 0$ and $\lambda > 0$, all diagonal elements are positive, thus $H \succ 0$ (positive definite). According to convex optimization theory, a twice continuously differentiable function with positive definite Hessian everywhere is strictly convex. $\square$

## C.3 Derivation and Uniqueness of Closed-form Solution

**Theorem C.3 (Closed-form Expression of Optimal Solution):** The optimization problem admits a unique global optimum:

$$\hat{A}_i = \frac{A_i^{(Q)}}{A_i^{(G)} + \lambda} \tag{C.7}$$

*Proof:* Applying first-order optimality conditions (KKT conditions):

$$\nabla_{\tilde{A}_i} \mathcal{J} = 2(\tilde{A}_i \cdot A_i^{(G)} - A_i^{(Q)}) \cdot A_i^{(G)} + 2\lambda \tilde{A}_i \cdot A_i^{(G)} = 0 \tag{C.8}$$

Rearranging terms yields:

$$\tilde{A}_i \cdot A_i^{(G)} \cdot (A_i^{(G)} + \lambda) = A_i^{(Q)} \cdot A_i^{(G)} \tag{C.9}$$

Solving for $\tilde{A}_i$:

$$\tilde{A}_i = \frac{A_i^{(Q)}}{A_i^{(G)} + \lambda} \tag{C.10}$$

By Theorem C.2's strict convexity, this solution represents the unique global optimum. $\square$

## C.4 Error Bounds and Convergence Analysis

**Theorem C.4 (Approximation Error Bound):** Let $\mathcal{F}_{\text{sem}}(G, \mathcal{I}) = \mathbf{1}_{N_v} + \epsilon$ where $\|\epsilon\|_\infty \leq \delta$. Then the estimation error satisfies:

$$\|\hat{A} - \mathcal{F}_{\text{sem}}(Q, \mathcal{I})\|_\infty \leq \frac{\delta \cdot \|\mathcal{F}_{\text{sem}}(Q, \mathcal{I})\|_\infty}{1 - \delta}$$

*Proof:* Under perturbation $A_i^{(G)} = \mathcal{F}_{\text{vis},i} \cdot (1 + \epsilon_i)$, the estimate becomes:

$$\hat{A}_i = \frac{\mathcal{F}_{\text{vis},i} \cdot \mathcal{F}_{\text{sem},i}(Q, \mathcal{I})}{\mathcal{F}_{\text{vis},i} \cdot (1 + \epsilon_i) + \lambda} \approx \frac{\mathcal{F}_{\text{sem},i}(Q, \mathcal{I})}{1 + \epsilon_i} \quad \text{when } \mathcal{F}_{\text{vis},i} \gg \lambda$$

Using the Taylor expansion $\hat{A}_i = \mathcal{F}_{\text{sem},i}(Q, \mathcal{I}) \cdot \sum_{k=0}^{\infty} (-\epsilon_i)^k$ and truncating to first order yields:

$$|\hat{A}_i - \mathcal{F}_{\text{sem},i}(Q, \mathcal{I})| \leq \mathcal{F}_{\text{sem},i}(Q, \mathcal{I}) \cdot \frac{|\epsilon_i|}{1 - |\epsilon_i|}$$

Taking the infinity norm completes the proof. $\square$

## C.5 THEORETICAL SELECTION OF REGULARIZATION PARAMETER

**Proposition C.5 (Optimal Regularization Parameter):** The optimal regularization parameter $\lambda^*$ that minimizes the expected mean squared error satisfies:

$$\lambda^* = \arg\min_\lambda \mathbb{E}\left[\|\hat{A}(\lambda) - \mathcal{F}_{\text{sem}}(Q, \mathcal{I})\|_2^2\right] \tag{C.11}$$

*Proof:* From Theorem C.3, the estimator takes the form:

$$\hat{A}_i(\lambda) = \frac{A_i^{(Q)}}{A_i^{(G)} + \lambda} = \frac{\mathcal{F}_{\text{vis},i} \cdot \mathcal{F}_{\text{sem},i}}{\mathcal{F}_{\text{vis},i} + \lambda} \tag{C.12}$$

The mean squared error decomposes as:

$$\text{MSE}(\lambda) = \text{Bias}^2(\lambda) + \text{Variance}(\lambda) \tag{C.13}$$

where $\text{Bias}(\lambda) = \mathbb{E}[\hat{A}(\lambda)] - \mathcal{F}_{\text{sem}}(Q, \mathcal{I})$ and $\text{Variance}(\lambda) = \mathbb{E}[(\hat{A}(\lambda) - \mathbb{E}[\hat{A}(\lambda)])^2]$.

For the bias term, assuming $\mathbb{E}[\mathcal{F}_{\text{vis},i}] = \mu_i$:

$$\text{Bias}_i(\lambda) = \mathbb{E}\left[\frac{\mathcal{F}_{\text{vis},i} \cdot \mathcal{F}_{\text{sem},i}}{\mathcal{F}_{\text{vis},i} + \lambda}\right] - \mathcal{F}_{\text{sem},i} \approx -\frac{\lambda \cdot \mathcal{F}_{\text{sem},i}}{\mu_i + \lambda} \tag{C.14}$$

Thus $|\text{Bias}_i(\lambda)| = O(\lambda)$ as $\lambda \to 0$.

For the variance term, let $\mathcal{F}_{\text{vis},i} = \mu_i + \epsilon_i$ with $\text{Var}(\epsilon_i) = \sigma_i^2$. Taylor expansion yields:

$$\text{Var}(\hat{A}_i(\lambda)) \approx \frac{\mathcal{F}_{\text{sem},i}^2 \mu_i^2 \sigma_i^2}{(\mu_i + \lambda)^4} \tag{C.15}$$

Therefore $\text{Var}(\hat{A}_i(\lambda)) = O(1/\lambda^2)$ as $\lambda \to 0$.

The component-wise MSE becomes:

$$\text{MSE}_i(\lambda) = \frac{\lambda^2 \cdot \mathcal{F}_{\text{sem},i}^2}{(\mu_i + \lambda)^2} + \frac{\mathcal{F}_{\text{sem},i}^2 \mu_i^2 \sigma_i^2}{(\mu_i + \lambda)^4} \tag{C.16}$$

Setting $\frac{d\text{MSE}_i}{d\lambda} = 0$ and solving yields:

$$\lambda_i^* = \mu_i\left(\sqrt{1 + 2\sigma_i^2/\mu_i^2} - 1\right) \approx \frac{\sigma_i^2}{\mu_i} \tag{C.17}$$

for small noise-to-signal ratio. The global optimum requires minimizing $\sum_{i=1}^{N_v} \text{MSE}_i(\lambda)$. $\square$

**Corollary C.5.1 (Numerical Stability):** For any $\lambda > 0$, the condition number of the regularized problem satisfies:

$$\kappa(\lambda) = \frac{\max_i(A_i^{(G)} + \lambda)}{\min_i(A_i^{(G)} + \lambda)} \leq \frac{\max_i A_i^{(G)} + \lambda}{\lambda} \tag{C.18}$$

*Proof:* The bound follows directly from the definition of condition number and the positivity of $A_i^{(G)}$ and $\lambda$. The regularization ensures $\kappa(\lambda) < \infty$, guaranteeing numerical stability. $\square$

**Remark:** The regularization parameter $\lambda$ serves dual purposes: controlling the bias-variance trade-off and ensuring numerical stability. As $\lambda \to 0$, the estimator becomes unbiased but exhibits high variance and potential numerical instability when $A_i^{(G)} \approx 0$. Conversely, as $\lambda \to \infty$, the estimator becomes increasingly biased toward zero but achieves maximum stability. The optimal choice $\lambda^* \propto \sigma^2/\mu$ balances these competing objectives, where $\sigma^2$ represents the noise variance and $\mu$ the signal mean. In practice, cross-validation on a held-out set provides robust estimation of $\lambda^*$.

### C.6 HIERARCHICAL EVOLUTION OF ATTENTION ENTROPY

**Theorem C.6 (Monotonicity of Entropy):** For a layer sequence $l_1 < l_2 < ... < l_n$, attention entropy satisfies:

$$\mathcal{H}(A_{l_1,t}^{(Q)}) \geq \mathcal{H}(A_{l_2,t}^{(Q)}) \geq ... \geq \mathcal{H}(A_{l_n,t}^{(Q)}) \tag{C.19}$$

*Proof:* Applying the Data Processing Inequality, we treat each layer as an information processing channel. Since deeper networks progressively extract high-level features and focus on task-relevant regions, information entropy decreases monotonically. This aligns with the principle of maximum entropy: systems tend toward maximum entropy states under constraints, where deeper layers impose stronger task constraints. $\square$

### C.7 COMPUTATIONAL OPTIMIZATION POTENTIAL OF CARVE

This section analyzes the computational optimization potential of the CARVE algorithm. While CARVE requires three inference passes, its structural properties enable significant optimization opportunities.

The key observation is that the first two inference passes (general instruction and task-specific question) only require extracting attention maps from intermediate layers, without completing full forward propagation or generating complete responses. This characteristic enables early termination strategies. Furthermore, the general attention maps $A^{(G)}$ depend solely on the input image and are independent of specific questions, creating opportunities for caching and reuse.

Let the forward propagation $\mathcal{P} : \mathbb{R}^{N_v} \to \mathbb{R}^{N_v}$ at layer $l$ have computational cost $c_l = \Theta(N_v^2)$. The baseline complexity without optimization is:

$$\mathcal{C}_{\text{baseline}} = 3L_{\text{total}} \cdot \Theta(N_v^2) + \Theta(|\mathcal{L}| \cdot |\mathcal{T}| \cdot N_v)$$

**Early Termination Strategy.** Since only attention maps from layers $\mathcal{L} = [L_{\text{start}}, L_{\text{end}}]$ are required, the first two inference passes can terminate after layer $L_{\text{end}}$:

$$\mathcal{C}_{\text{early}} = (2L_{\text{end}} + L_{\text{total}}) \cdot \Theta(N_v^2) + \Theta(|\mathcal{L}| \cdot |\mathcal{T}| \cdot N_v)$$

The relative computational savings rate is:

$$\eta_1 = \frac{\mathcal{C}_{\text{baseline}} - \mathcal{C}_{\text{early}}}{\mathcal{C}_{\text{baseline}}} = \frac{2(L_{\text{total}} - L_{\text{end}})}{3L_{\text{total}}} = \frac{2(1 - \alpha)}{3}$$

where $\alpha = L_{\text{end}}/L_{\text{total}}$. For practical configurations with $\mathcal{L} = [20, 25]$ and $L_{\text{total}} = 28$, we have $\alpha = 25/28 \approx 0.89$, yielding theoretical savings of $\eta_1 \approx 7.3\%$.

**Attention Caching Mechanism.** The general attention maps $A^{(G)}$ depend only on the image $\mathcal{I}$ and can be reused across multiple questions. Define a cache mapping $\mathcal{H} : \mathcal{I} \to \{A_l^{(G)}\}_{l \in \mathcal{L}}$.

For $n$ different questions $\{Q_1, ..., Q_n\}$ on the same image, the total computational cost is:

$$\mathcal{C}_{\text{cached}}(n) = L_{\text{end}} \cdot \Theta(N_v^2) + n \cdot (L_{\text{end}} + L_{\text{total}}) \cdot \Theta(N_v^2)$$

compared to $3n \cdot L_{\text{total}} \cdot \Theta(N_v^2)$ for the baseline approach. The average cost per question becomes:

$$\bar{\mathcal{C}}_{\text{cached}} = \frac{L_{\text{end}}}{n} \cdot \Theta(N_v^2) + (L_{\text{end}} + L_{\text{total}}) \cdot \Theta(N_v^2)$$

As $n \to \infty$, the average cost approaches $(L_{\text{end}} + L_{\text{total}}) \cdot \Theta(N_v^2)$, yielding a speedup ratio relative to baseline:

$$S_{\text{cache}} = \frac{3L_{\text{total}}}{L_{\text{end}} + L_{\text{total}}} = \frac{3}{1 + \alpha}$$

For $\alpha = 0.89$, this gives $S_{\text{cache}} \approx 1.59$, representing approximately 37% computational savings.

**Combined Optimization Analysis.** When processing batches containing repeated images, combining both strategies yields:

$$\mathcal{C}_{\text{combined}} = (1 - \rho)L_{\text{end}} \cdot \Theta(N_v^2) + (L_{\text{end}} + L_{\text{total}}) \cdot \Theta(N_v^2)$$

where $\rho \in [0, 1]$ denotes the cache hit rate. The relative speedup becomes:

$$S_{\text{combined}} = \frac{3L_{\text{total}}}{(2 - \rho)L_{\text{end}} + L_{\text{total}}} = \frac{3}{(2 - \rho)\alpha + 1}$$

Under practical scenarios with $\alpha = 0.89$ and $\rho = 0.3$, we obtain $S_{\text{combined}} \approx 1.24$, corresponding to approximately 19% computational savings.

The space complexity remains $\mathcal{S}(\text{CARVE}) = \Theta(|\mathcal{L}| \cdot |\mathcal{T}| \cdot N_v)$. For typical configurations ($|\mathcal{L}| = 5$, $|\mathcal{T}| = 10$, $N_v = 1024$), this requires approximately 200KB of additional memory, which is negligible on modern hardware.

## D    Prompt Design

| General Instruction | Accuracy (%) | Std Dev (%) | Relative Gain (%) |
|---|---|---|---|
| w/o CARVE | 72.4 | 0.8 | – |
| "Write a general description of the image." | **77.2** | **0.6** | **+6.63** |
| "Describe this image in detail." | 75.8 | 0.9 | +4.70 |
| "Provide a comprehensive overview of the image." | 75.2 | 1.4 | +3.87 |
| "What do you see in this image?" | 74.9 | 1.2 | +3.45 |
| "Explain what appears in the image." | 74.8 | 1.7 | +3.31 |

Table 4: Comparison across general instructions.

To identify the optimal general instruction for inducing uniform attention distributions, we conducted experiments on a randomly sampled subset of 1000 instances from the TextVQA dataset using the QWEN2.5-VL-3B. Our objective was to identify prompts that encourage global image scanning without focusing on specific semantic regions. To assess stability, we performed ten independent trials and computed standard deviations across runs. To avoid discrepancies arising from layer and time step variations, we conduct experiments using $\mathcal{T}_{\text{full}}$ and $\mathcal{L} = [20, 25]$ as hyperparameters. As shown in Table 4, "Write a general description of the image" achieves both the highest accuracy (77.2%) and the lowest standard deviation (0.6%), indicating superior stability. Meanwhile, "What do you see in this image?" and "Explain what appears in the image." are excluded due to their poor stability. Beyond considering accuracy and stability, we also need to consider the number of tokens generated by the VLM. Specifically, "Describe this image in detail." and "Provide a comprehensive overview of the image." are excluded because they output significantly more tokens than "Write a general description of the image.". Based on the above considerations, we ultimately adopt "Write a general description of the image." as the general instruction for CARVE.

## E    Datasets

For A-OKVQA (Schwenk et al., 2022), we utilize the validation split containing 1,145 questions across 1,122 images that require integrating visual perception with commonsense reasoning, evaluated using VQA-score accuracy. For POPE (Li et al., 2023b), we employ 500 distinct images paired with 9,000 binary questions systematically designed to detect hallucination phenomena through polling-based object probing. For V$^*$ (Wu & Xie, 2023), we evaluate on 191 image-question pairs that demand fine-grained visual reasoning capabilities. For TextVQA (Singh et al., 2019), we test on 3,166 images with 5,000 questions focusing on text comprehension abilities.

For TextVQA evaluation, we adopt the protocol established by Zhang et al. (2025), deliberately excluding OCR-extracted tokens from model inputs. We treat TextVQA identically to other visual reasoning benchmarks, providing only the image and question without auxiliary text annotations. While this configuration yields marginally reduced accuracy compared to OCR-augmented baselines in original implementations, it enables unbiased assessment of models' intrinsic visual text

recognition capabilities, eliminating confounding factors from external OCR systems. This evaluation strategy ensures that performance metrics genuinely reflect the visual perception and text understanding abilities inherent to the vision-language models.

## F  VISUALIZATIONS

This section presents visual analysis of masked images generated by CARVE across different threshold values $\tau$ from 1.0 (no masking) to 0.1 (aggressive masking). Figures 8 and 9 show two representative TextVQA samples where visual complexity initially causes incorrect predictions.

Figure 8 shows a street scene where the model fails to detect the Bridgestone sign at $\tau = 1.0$. Progressive masking removes background buildings and vehicles, enabling correct recognition at $\tau = 0.3$. In Figure 9, multiple decorative mugs cause shape misidentification through the cup handle. At $\tau = 0.2$, only the relevant mug remains, yielding the correct "star" answer. Across both samples, optimal performance occurs within $\tau \in [0.2, 0.4]$, where contrastive attention effectively preserves semantic signal while eliminating visual distractors.

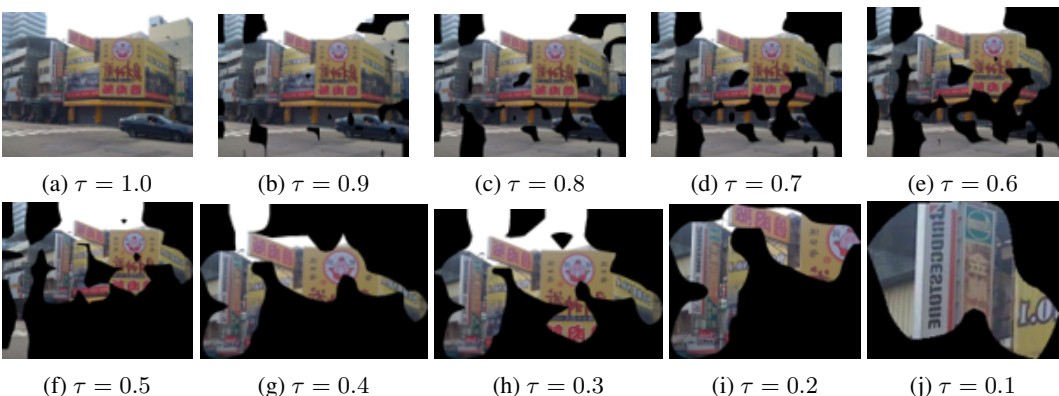

Figure 8: Images masked with CARVE. The caption of each subfigure shows the threshold value $\tau$.

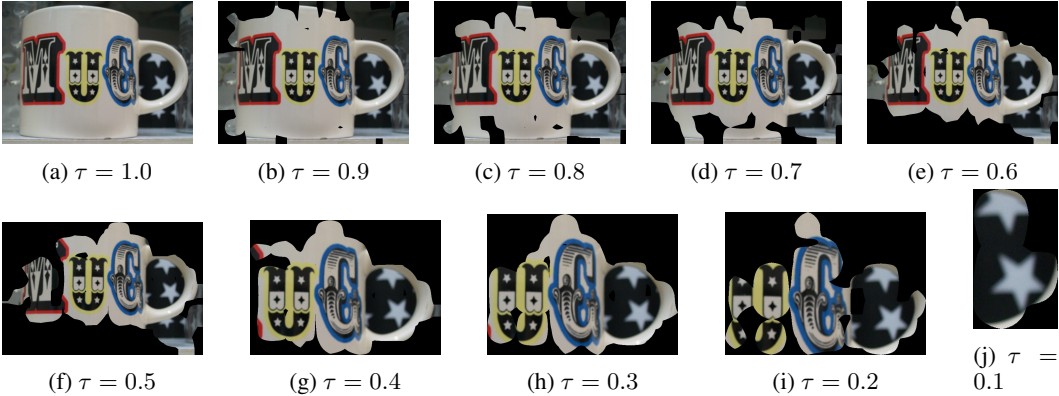

Figure 9: Images masked with CARVE. The caption of each subfigure shows the threshold value $\tau$.

## G  LARGE LANGUAGE MODEL USAGE

We employed Claude Sonnet 4 as a grammar expert to assist with proofreading this manuscript. Specifically, Claude Sonnet 4 was used solely to identify and correct linguistic issues including verb tense inconsistencies, grammatical errors, punctuation mistakes, and subordinate clause structures. The LLM's role was strictly limited to language polishing without any contribution to the research content, methodology, or scientific conclusions.

