# OpenReview forum: "Focusing by Contrastive Attention: Enhancing VLMs' Visual Reasoning"
_ICLR.cc/2026/Conference — ICLR 2026 Conference Withdrawn Submission_

### Official Review · Reviewer_EEGZ · 2025-10-28

**Soundness:** 3
**Presentation:** 3
**Contribution:** 3
**Rating:** 4
**Confidence:** 3

**Summary:**

This paper presents an empirical analysis showing a positive correlation between visual complexity and attention entropy, and a negative correlation between attention entropy and visual reasoning accuracy. Building on these findings, it introduces a training-free method, CARVE, which contrasts attention maps from a “general instruction” and a “task-specific question,” normalizes them to obtain semantically relevant attention, and then performs pixel-level masking, cropping, and magnification to suppress visual noise and focus on task-relevant regions. The paper evaluates CARVE on Qwen2.5-VL and LLaVA models across several typical VQA benchmarks, reporting substantial gains.

**Strengths:**

1. The paper quantifies visual complexity via texture density and hue diversity, measures attention dispersion with Shannon entropy, and demonstrates a correlation chain linking complexity, attention entropy, and accuracy. A longitudinal inter-layer analysis further shows that deeper layers exhibit more concentrated attention but increased variance, reinforcing the method’s motivation.

2. The approach requires only a single forward pass to extract two attention maps and take their contrast; no external segmenter or additional training is needed.

3. Evidence spans models and datasets, includes ablations across layers and time steps, sensitivity studies on thresholds and the number of regions, and comparisons against SAM/YOLO/CLIP/ViCrop, which together provide a well-rounded empirical case.

**Weaknesses:**

1. The paper instantiates complexity through two proxies: Canny edge detection and Hue distribution, but offers limited references and few independent validation studies. Moreover, in settings with comparable edge density and color statistics (e.g., Table VQA benchmarks), attributing failures to attention dispersion driven by these proxies remains debatable.

2. In the exploration phase, the attention entropy H is computed on contrasted attention by default, yet the design choice is not thoroughly analysed. In the method, contrasted attention is further treated as an effective estimator of semantic localization to generate masks, but there is no alignment against external localization ground truth, nor a direct comparison with the native attention readily available from standard models.

3. The use of contrasted attention assumes that attention under the general instruction primarily reflects task-agnostic visual factors like noise or background. The exploration section does not directly test or substantiate this premise.

**Questions:**

NA

---

### Official Review · Reviewer_whRJ · 2025-10-29

**Soundness:** 2
**Presentation:** 3
**Contribution:** 2
**Rating:** 4
**Confidence:** 4

**Summary:**

This paper defined a “visual complexity” concept including texture and color complexity, then explored how they affect Vision-Language Models' (VLMs) attention and discovered some phenomena. Building upon the findings, the authors proposed a training-free method called CARVE. Concretely, CARVE generated contrastive attention maps to mask visual noise, crop relevant regions, and magnify them for improved inference, thereby improving visual reasoning. Experimental results exhibited improvements across multiple VLMs and benchmarks, demonstrating the effectiveness of the proposed method.

**Strengths:**

1. This paper analyzed VLMs from perspectives of visual complexity and attention entropy, finding several intuitive phenomena. With the findings, the authors presented the CARVE method to enhance the visual reasoning capability of VLMs.

2. The proposed CARVE is training-free, the empirical results demonstrated its effectiveness on different VLMs and test sets, and model scales range from 3B to 13B, making it feasible for various visual reasoning scenarios.

**Weaknesses:**

1. Numerous previous studies have already investigated VLMs’ attention mechanism and its relationship with hallucination or visual reasoning, such as [Devils, CVPR 2025], [EAH, EMNLP 2025], [Farsight, CVPR 2025], [TAME, ICLR 2025], [FastV, ECCV 2024], [Clearsight, CVPR 2025], and [SEE WHAT YOU ARE TOLD, ICLR 2025]. More importantly, there have been long Chain-of-Thought reasoning VLMs since OpenAI o1 and DeepSeek-R1(-0528), such as VLM-R^3, LlamaV-o1, Vision-R1, and Visual-O1, which are all of strong visual reasoning capabilities on most of benchmarks. It is necessary for the authors to clarify the exclusive contribution of the proposed method, explain limitations of the prior works, and explicitly distinguish their proposed method from them.

2. The “visual complexity” concept based solely on "texture and color dimensions" was poorly justified, there is no validation that these dimensions can capture relevant aspects, or discussion of why other visual aspects, such as object density, or lighting variations were ignored.

3. The paper claimed to “prove” that contrastive attention scores enable decomposition of visual signals into semantic and noise components, but this claim lacked enough theoretical proofs. For example, the core theoretical claim in Definition 1 that attention decomposes as Eq. 4.1 lacked rigorous justification, and Appendix A.2 didn’t interpret that clearly. The authors assumed this decomposition exists, but didn't establish why the specific multiplicative decomposition should hold neither from theoretical proofs nor empirical validations. Apart from that, Eq. 4.3 and Eq. 4.4 are more like heuristic designs rather than theoretically grounded.

4. The experimental evaluation of this work was limited. Results are primarily shown only for TextVQA dataset, with no comprehensive testing on standard benchmarks like GQA or ScienceQA. Also, the paper requires proper baselines to demonstrate the superiority of the proposed CRAVE, more comparisons need to be conducted, such as such as [Devils, CVPR 2025], [EAH, EMNLP 2025], [Farsight, CVPR 2025], [TAME, ICLR 2025], [FastV, ECCV 2024], [Clearsight, CVPR 2025], and [SEE WHAT YOU ARE TOLD, ICLR 2025].

5. The Abstract claimed “75% improvement” is misleading, even this paper adopted ratio metric to scale up the limited improvements, the results are up to 71.83% at the old Llava1.5-7B. Additionally, there was no statistical significance testing for empirical results, and no confidence or variance intervals, making the reported results random and unreliable.

**Questions:**

1. Given the long Chain-of-Thought reasoning methods, such as OpenAI o1 and DeepSeek-R1(-0528) distilled LLMs and VLMs (e.g., LLaVA-o1, LlamaV-o1, VLM-R3), could the current short CoT models outperform these o1-like VLMs? If not, what is the meaningfulness of this research?

2. Why could general queries capture noise while specific queries capture semantic information, is this assumption universally valid or theoretically grounded? And what if both general and specific queries produce similar attention patterns?

3. All experiments were conducted with greedy decoding, how would the CRAVE perform under a common sampling setting?

4. How could the proposed CARVE handle the circumstance where the correct reasoning demands integrating information from multiple disparate regions of an input image? Also, this work assumed general instructions induce uniform semantic signals, but does that always hold? Some images might naturally focus on certain regions regardless of instructions.

---

> ### Author Response · Authors · 2025-11-13
> **Response to Reviewer whRJ**
>
> Thank you very much for taking the time to review and for your support. We try our best to address your questions as follows.
>
> > Given the long Chain-of-Thought reasoning methods, such as OpenAI o1 and DeepSeek-R1(-0528) distilled LLMs and VLMs (e.g., LLaVA-o1, LlamaV-o1, VLM-R3), could the current short CoT models outperform these o1-like VLMs? If not, what is the meaningfulness of this research?
>
> Thank you for this insightful question. CARVE does not compete with o1-like reasoning VLMs; rather, it complements them. A core advantage of CARVE is its broad applicability—it can be applied to any Transformer-based VLM to enhance visual perception capabilities. Specifically, CARVE improves visual understanding by contrasting attention maps derived from general and task-specific instructions to identify and suppress visual noise, thereby enabling the model to focus on semantically relevant regions. Since o1-like reasoning VLMs share the same fundamental Transformer architecture as short CoT models, CARVE can similarly enhance their visual perception abilities.
>
> Due to budget constraints, we cannot conduct large-scale testing on o1-like reasoning VLMs. To provide a comprehensive evaluation, we randomly sampled 100 image-question pairs from TextVQA for a case study. We extracted attention maps from Qwen2.5-VL-3B under both general and specific instructions, computed the contrastive attention map, and applied the resulting mask to the input images before feeding them to Qwen3-VL-235B-A22B-Thinking [1] (currently the state-of-the-art o1-like visual reasoning model). We chose Qwen2.5-VL-3B for attention extraction to reduce computational costs, leveraging the attention consistency across the Qwen model family. Results show that accuracy improved from 92% (vanilla) to 98%, representing a 6.5% relative improvement.
>
> > Why could general queries capture noise while specific queries capture semantic information, is this assumption universally valid or theoretically grounded? And what if both general and specific queries produce similar attention patterns?
>
> Thank you for this thoughtful question. To address it, consider the following scenario: You are presented with an image of a puppy on a lawn. If asked to "Write a general description of the image," you would scan the entire image, resulting in dispersed attention. However, if subsequently asked "What is the color of the puppy?", your attention would concentrate on the puppy, though the lawn background might still introduce visual interference. In this second query, the lawn constitutes visual noise. Since the first query also contains visual noise, contrasting the attention maps from both queries allows us to eliminate the lawn's interference. This assumption holds universally across diverse visual scenarios. Importantly, if both queries produce similar attention patterns, CARVE does not degrade performance; instead, it enhances the model's focus on the target object, thereby improving effectiveness.
>
> > All experiments were conducted with greedy decoding, how would the CRAVE perform under a common sampling setting?
>
> In general visual reasoning tasks, researchers commonly employ low temperature settings or greedy decoding to achieve stable reasoning performance and consistent evaluation metrics. Therefore, we adopted the same experimental settings to ensure fair comparison.
>
> To address your question, we randomly sampled 100 image-question pairs from TextVQA for a case study. We conducted three trials with Qwen2.5-VL-3B using decoding temperatures of 0.3, 0.6, and 1.0, and averaged the results. The accuracies were 76.3%, 77.6%, and 77.9%, respectively.
>
>
> [1] https://huggingface.co/Qwen/Qwen3-VL-235B-A22B-Thinking

---

> ### Author Response · Authors · 2025-11-13
> **Response to Reviewer whRJ**
>
> > How could the proposed CARVE handle the circumstance where the correct reasoning demands integrating information from multiple disparate regions of an input image? Also, this work assumed general instructions induce uniform semantic signals, but does that always hold? Some images might naturally focus on certain regions regardless of instructions.
>
> Thank you for raising this important concern. CARVE's ability to handle multiple disparate regions is inherently limited by its design philosophy of enhancing local detail perception through focused cropping.
>
> Our multi-scale bounding box selection strategy attempts to mitigate this limitation by selecting regions that contain the visual subject as completely as possible. The parameter K (maximum number of kept regions) provides some flexibility—increasing K allows the method to preserve multiple important regions. However, when a question requires integrating information from visual subjects that are both small and spatially distant, CARVE can only effectively enhance perception of the region the model considers most important based on its internal attention distribution.
>
> We conducted exploratory tests on the blink_depth task from the Blink Benchmark [2], which provides an illuminating case study of CARVE's design trade-offs. This dataset contains photo-question pairs where each photo includes two red circles, and the question asks which circle is closer to the camera. (A sample can be viewed at https://zeyofu.github.io/blink/static/images/teaser.png, where the "Related depth" example in the upper left corner illustrates this task.)
>
> On this task, CARVE precisely crops out the red circles as intended. However, this focused cropping reduces accuracy because determining relative depth requires analyzing background visual cues—buildings, roads, and other contextual elements that indicate spatial relationships. By masking the background to enhance attention on local details (the circles themselves), CARVE inadvertently removes the contextual information essential for spatial reasoning.
>
> This case nicely illustrates CARVE's fundamental design trade-off: by removing background context to enhance local detail perception, the method sacrifices global spatial reasoning capabilities. This finding confirms that CARVE is most effective for tasks requiring fine-grained perception of local visual details—such as OCR, small object recognition, and detailed visual question answering—rather than tasks requiring holistic spatial relationship reasoning or multi-object comparative analysis that depend heavily on background context.
>
> We will add explicit discussion of this limitation and the scope of CARVE's applicability in the revised version. Thank you for this insightful feedback that helps clarify when and how CARVE should be applied!
>
> [2] BLINK: Multimodal Large Language Models Can See but Not Perceive

---

> ### Comment · Reviewer_whRJ · 2025-11-27
>
> Thank you for the responses! After reviewing the rebuttals and the manuscript (not been revised), I notice my concerns are still hanging there.
> - The authors mentioned that the proposed CARVE can complement the openai-o1/deepseek-r1-like VLMs, however, there are no empirical results to support such a claim, even on a limited output budget.
> - There is no discussion on the differences between the presented CARVE and the closely related work, severely undermining the novelty and contribution of this work.
> - Without extra experimental results, the current explanations or theoretical derivations for the discrete focus, regional pixels caption cannot be interpreted or understood.
>
> Given the above, I decide to maintain my reject rating.

---

### Official Review · Reviewer_pgMe · 2025-10-31

**Soundness:** 3
**Presentation:** 3
**Contribution:** 3
**Rating:** 4
**Confidence:** 4

**Summary:**

The paper studies why VLMs fail to “focus” in complex scenes and proposes a training-free enhancement, CARVE (Contrastive Attention Refinement for Visual Enhancement). Empirically, the authors show that (i) visual complexity (texture/color) correlates with higher attention entropy and lower accuracy; (ii) attention sharpens from shallow to deep layers; and (iii) contrasting attention from a general instruction vs. the task question yields a pixel-level mask that suppresses visual noise, improving VQA-style benchmarks across LLaVA-1.5 and Qwen2.5-VL models.

**Strengths:**

1. A simple inference-time refinement method for VQA reasoning tasks that improves accuracy without additional training or heavy compute.
2. Plug-and-play method applicable to transformer-based VLMs.
3. Clear explanation of the problem.

**Weaknesses:**

1. Potential degradation on reasoning tasks beyond VQA, especially those requiring global context.
2. Limited evaluation scope: only four datasets were used for evaluation. Given that no training is involved, this is relatively small for VLM evaluation and for demonstrating the generalizability of the results.
3. Relatively small gains given ~3× inference passes. The paper mentions early termination to reduce two of the three passes but the total overhead remains underexplored.

**Questions:**

1. Following the weaknesses above, I am a bit concerned about the generalizability of the proposed method, beyond VQA reasoning tasks that rely on local context. Could you please elaborate on this? I would be interested in seeing negative/neutral cases where CARVE hurts or offers no benefit (e.g., wrong region kept; too small K; OCR-like questions).
2. Related to the above question, could you please elaborate on how the model would perform on more general reasoning tasks, such as image captioning? In those cases, I would guess that the general and the question attention masks become very similar, which could be a failure point for the proposed approach.
3. How much additional memory and compute does the method require at inference compared to the raw model without CARVE? According to the paper, the number of passes is tripled (once for the general instruction, once for the question, and once for the question with the masked image).
4. What does “accuracy” exactly mean in Table 3, and how is it computed for models like SAM and YOLO?
5. In Figure 5-b, both performance and entropy are plotted on a single y-axis; it is unclear what the y-axis represents and what the exact performance values are for each point. Could you please clarify Figure 5-b in more detail?

---

### Official Review · Reviewer_fSTn · 2025-10-31

**Soundness:** 3
**Presentation:** 3
**Contribution:** 2
**Rating:** 4
**Confidence:** 3

**Summary:**

This paper investigates the degradation of VLMs performance in visually complex scenes by providing a quantitative analysis that establishes a clear link between visual complexity (defined by texture and color), increased attention entropy, and reduced model accuracy. Based on this finding, the paper proposes CARVE, a training-free method to improve VLM reasoning. The core idea is to contrast the attention maps generated from a task-specific question against those from a general, descriptive prompt. This contrast is designed to isolate task-relevant "semantic signals" from task-agnostic "visual noise." Extensive experiments on several VLMs, demonstrate consistent performance improvements across multiple benchmarks.

**Strengths:**

1. While the phenomenon that "complex visual information impairs performance" is intuitive, the authors' contribution lies in rigorously and quantitatively analyzing this relationship. By defining concrete metrics for visual complexity and linking them empirically to attention entropy and downstream task performance (Figs. 4 & 5), the paper provides a solid, data-driven foundation for its claims. This analysis transforms an intuitive observation into a measurable scientific insight, which is a significant strength.

2. Based on this finding, This paper demonstrates clear effectiveness as a training-free enhancement. The ability to improve performance on a strong, recent model like Qwen2.5-VL  is particularly impressive and suggests that even advanced VLMs suffer from attention dispersion and can benefit from this approach. The substantial performance gains observed on weaker models (e.g., up to 75% on LLaVA-1.5-13B) further underscore the utility of CARVE as a mechanism to compensate for inherent model limitations in focusing ability.

**Weaknesses:**

1. A primary weakness of the proposed method is its sensitivity to hyperparameters, which is a significant concern for a training-free approach that aims for broad applicability. Figure 7 clearly shows that the performance of CARVE is highly dependent on the choice of the top-p threshold (p) and the maximum number of kept regions (K). For a method to be truly "plug-and-play," it should be robust to such choices or provide a principled, automatic way to set them. The current need for manual tuning for optimal results somewhat undermines the convenience of its training-free nature.

2. It is hoped that more models, especially more experiments on complex visual scene benchmarks, will be conducted to further enhance the effectiveness and persuasiveness of the method proposed in this paper.

If the robustness of the method and the performance gains on more benchmarks can be further demonstrated, I am willing to increase the score.

**Questions:**

See weaknesses.

---

> ### Author Response · Authors · 2025-11-13
> **Response to Reviewer fSTn**
>
> Thank you for your thoughtful review. We address your concerns below:
>
> > A primary weakness of the proposed method is its sensitivity to hyperparameters, which is a significant concern for a training-free approach that aims for broad applicability. Figure 7 clearly shows that the performance of CARVE is highly dependent on the choice of the top-p threshold (p) and the maximum number of kept regions (K). For a method to be truly "plug-and-play," it should be robust to such choices or provide a principled, automatic way to set them. The current need for manual tuning for optimal results somewhat undermines the convenience of its training-free nature.
>
> Thank you for this important point. We respectfully argue that adjustable parameters represent a **design advantage rather than a limitation**, enabling users to tailor CARVE to different scenarios—traffic monitoring can employ smaller K and lower p for aggressive focusing, while document understanding can use larger K and higher p to preserve context.
>
> As detailed in **Section 5.2 (lines 460-465)** and **Figure 7**, our sensitivity analyses on TextVQA demonstrate strong robustness. For Qwen2.5-VL-7B with K=3, performance peaks at p=0.4 (82.1%) and remains consistently high across p∈[0.2, 0.6] (78.0%-82.1%), substantially exceeding baseline (75.0%). With p=0.4 fixed, accuracy remains robust for K∈[2, 4] (80.0%-82.1%). Even at boundary settings (p=1.0 or K=1), CARVE gracefully degrades to baseline without harmful effects.
>
> These results show CARVE maintains consistent improvements across broad parameter ranges without requiring precise tuning. As a training-free method, CARVE prioritizes **practical utility and user control**, allowing practitioners to make informed trade-offs based on application requirements.
>
> ---
>
> > It is hoped that more models, especially more experiments on complex visual scene benchmarks, will be conducted to further enhance the effectiveness and persuasiveness of the method proposed in this paper.
>
> Thank you for this suggestion. Table 2 evaluates CARVE across four models (3B to 13B) from two architectures. For Qwen2.5-VL, TextVQA improvements reach 4.8% (3B) and 8.9% (7B). For LLaVA-1.5, improvements reach 17.57% (7B) and 21.76% (13B). This cross-architecture validation demonstrates CARVE's broad generalizability without architecture-specific adaptations.
>
> Regarding complex visual scenarios, CARVE is designed for fine-grained local perception (OCR, small object recognition, detailed VQA) rather than holistic spatial reasoning. To illustrate this boundary, we tested the blink_depth task from Blink Benchmark [1] (https://zeyofu.github.io/blink/static/images/teaser.png, "Related depth" example), which requires comparing relative depth between two red circles. CARVE precisely crops the circles but reduces accuracy because depth judgment requires background cues—buildings, roads, perspective lines. By masking background to enhance local perception, CARVE removes contextual information essential for spatial reasoning.
>
> This demonstrates CARVE's fundamental design trade-off: removing background context enhances local detail perception but sacrifices global spatial reasoning. This validates CARVE's effectiveness for target scenarios while clarifying principled limitations.
>
> CARVE's plug-and-play universality requires no architecture-specific modifications or dataset-specific tuning, contrasting with training-based methods demanding extensive GPU resources.
>
> We will incorporate blink_depth analysis in the Appendix, providing clear guidance on optimal deployment scenarios.
>
> **Reference:**
> [1] BLINK: Multimodal Large Language Models Can See but Not Perceive.

---

### Official Review · Reviewer_WNpL · 2025-11-09

**Soundness:** 3
**Presentation:** 4
**Contribution:** 3
**Rating:** 4
**Confidence:** 3

**Summary:**

The paper introduces CARVE, a training-free method that compares attention from a general instruction and with the task question, to create a pixel-level mask that removes visual noise and improves VLM reasoning focus on the core part. It shows that visual complexity increases attention entropy, and higher entropy reduces accuracy. They conduct experiments to show consistent gains using their attention entropy based training-free methods across models and datasets, with ablations on time steps, layers, and mask parameters.

**Strengths:**

1. CARVE shows consistent improvements over the original models and external-tool baselines;
2. The paper formalizes attention-entropy clearly, shows complexity -> entropy and entropy -> accuracy trends, and provides a closed-form contrasted attention used for masking.
3. Abalation results are through across time steps and layers, plus sensitivity to top-p and region count K, with practical guidance. They also compare with SAM/YOLO/CLIP/ViCrop and gradient/attention variants.

**Weaknesses:**

Refer to Questions section.

**Questions:**

1. For SAM/YOLO/CLIP/ViCrop, were thresholds, confidence cutoffs, and post-processing tuned to near-optimal for each dataset & model  to ensure a fair comparison?
2. The paper needs an error case analysis to clarify the limits of the method and guide usage. Does it still work for tasks with global-context or multi-object queries, or other VQA tasks involving more reasoning?
3. Have you considered in-attention sharpening (no image masking) that directly sharpens the difference between question and general attention? For example: compute (A' <- (A^Q)/(A^G+\lambda)) and use softmax(log A'/t) to renormalize or sharpen the cross-attention weights. It could work as a baseline very direct to the motivation.

If the authors answer my questions well and make these clarifications, I am happy to raise my review score.

---

> ### Author Response · Authors · 2025-11-13
> **Response to Reviewer WNpL**
>
> Thank you very much for taking the time to review and for your support. We try our best to address your questions as follows.
>
>
> > For SAM/YOLO/CLIP/ViCrop, were thresholds, confidence cutoffs, and post-processing tuned to near-optimal for each dataset & model to ensure a fair comparison?
>
> Thank you for this important concern. We used the recommended default configurations for all external tools without dataset-specific hyperparameter tuning, following the settings in reference [1].
>
> Below are the complete experimental settings for SAM/YOLO/CLIP/ViCrop methods.
>
> **SAM:** We utilized pretrained SAM [2] with automatic mask generation mode. Processing steps: (1) SAM generates extensive segmentation masks covering all objects and parts; (2) each mask is converted to its smallest covering bounding box; (3) CLIP [3] computes semantic similarity between each box region and the question; (4) the highest-scoring box is selected and resized to a square format to avoid deformation.
>
> **YOLO:** We employed YOLOv8 [4] pretrained on COCO [5] with confidence threshold 0.25 (recommended default). Processing steps: (1) YOLO detects all bounding boxes with confidence >0.25, filtering regions without salient objects; (2) each box is cropped and its CLIP similarity to the question is computed; (3) the highest-scoring box is selected and adjusted to square format.
>
> **CLIP:** We applied progressive refinement with pretrained CLIP [3]. Parameters: cropping ratio 0.9, four directions (top, bottom, left, right), 20 iterations. Process: Starting with the full image, each iteration generates four overlapping crops by cropping from each direction at ratio 0.9. CLIP selects the highest-scoring crop for the next iteration. After 20 iterations, the final crop with highest similarity is adjusted to square format.
>
> All cropped regions are resized to match each VLM's input resolution, then provided alongside the original image.
>
> As shown in Table 7:
>
> | Method | Original | SAM | YOLO | CLIP | ViCrop (rel-att) | ViCrop (grad-att) | ViCrop (pure-grad) | CARVE |
> |--------|----------|-----|------|------|------------------|-------------------|-------------------|-------|
> | Accuracy (%) | 47.80 | 49.42 | 48.84 | 48.55 | 55.17 | 56.06 | 51.67 | **58.2** |
>
> **CARVE consistently outperforms external tools across all metrics, validating the effectiveness of utilizing the VLM's internal knowledge for question-dependent localization. We will add the above experimental settings to the appendix in the next version.**
>
> -----
>
> > The paper needs an error case analysis to clarify the limits of the method and guide usage. Does it still work for tasks with global-context or multi-object queries, or other VQA tasks involving more reasoning?
>
> Thank you for this valuable suggestion. We conducted exploratory tests on the blink_depth task from the Blink Benchmark [6], which provides an illuminating case study of CARVE's design trade-offs. This dataset contains photo-question pairs where each photo includes two red circles, and the question asks which circle is closer to the camera. (A sample can be viewed at https://zeyofu.github.io/blink/static/images/teaser.png, where the "Related depth" example in the upper left corner illustrates this task.)
>
> On this task, CARVE precisely crops out the red circles as intended. However, this focused cropping reduces accuracy because determining relative depth requires analyzing background visual cues—buildings, roads, and other contextual elements that indicate spatial relationships. By masking the background to enhance attention on local details (the circles themselves), CARVE inadvertently removes the contextual information essential for spatial reasoning.
>
> This case nicely illustrates CARVE's fundamental design trade-off: by removing background context to enhance local detail perception, the method sacrifices global spatial reasoning capabilities. **This finding confirms that CARVE is most effective for tasks requiring fine-grained perception of local visual details—such as OCR, small object recognition, and detailed visual question answering—rather than tasks requiring holistic spatial relationship reasoning or multi-object comparative analysis that depend heavily on background context.**
>
> We will add explicit discussion of this limitation and the scope of CARVE's applicability in the revised version. Thank you for this insightful feedback that helps clarify when and how CARVE should be applied!
>
> ---
>
> **Reference:**
>
> [1] MLLMs Know Where to Look: Training-free Perception of Small Visual Details with Multimodal LLMs
>
> [2] Segment Anything
>
> [3] Learning Transferable Visual Models From Natural Language Supervision
>
> [4] Real-Time Flying Object Detection with YOLOv8
>
> [5] Microsoft COCO: Common Objects in Context
>
> [6] BLINK: Multimodal Large Language Models Can See but Not Perceive

---

> ### Author Response · Authors · 2025-11-13
> **Response to Reviewer WNpL**
>
> > Have you considered in-attention sharpening (no image masking) that directly sharpens the difference between question and general attention? For example: compute (A' <- (A^Q)/(A^G+λ)) and use softmax(log A'/t) to renormalize or sharpen the cross-attention weights. It could work as a baseline very direct to the motivation.
>
> Thank you for this very insightful alternative suggestion, which indeed aligns with CARVE's design philosophy. In fact, as you suggested, in the early stages of this work, we attempted to improve VLM's attention distribution by modifying the model's internal representations. However, due to the architectural differences across various vision models, this approach exhibited high variance across different models and datasets, resulting in poor generalizability.
>
> In contrast, CARVE does not alter the model's internal parameters or internal representations. Instead, by removing noisy regions and magnifying key regions, it provides higher-quality visual input to the vision encoder, thereby achieving stronger universality and generalizability. This design choice allows CARVE to work as a plug-and-play enhancement across different VLM architectures without requiring architecture-specific adaptations or risk of destabilizing the pretrained attention mechanisms.
>
> We appreciate this suggestion and believe that exploring hybrid approaches (combining in-attention sharpening with physical cropping) could be a promising direction for future work.

---

### Note · Authors · 2026-01-01

I have read and agree with the venue's withdrawal policy on behalf of myself and my co-authors.